

**Title:** The relative importance of photodegradation and biodegradation of terrestrially derived
dissolved organic carbon across four lakes of differing trophic status
**Authors:** Christopher M Dempsey[1][*], Jennifer A Brentrup[2], Sarah Magyan[1], Lesley B Knoll[3],
Hilary M Swain[4], Evelyn E Gaiser[5], Donald P Morris[6], Michael T Ganger[1], and Craig E
Williamson[7]
**Author Affiliations**:
[1]Gannon University, Biology Department, Erie, PA, USA
[2]University of Vermont, Rubenstein Ecosystem Science Laboratory, Burlington, VT, USA
[3]University of Minnesota, Itasca Biological Station and Laboratories, Lake Itasca, MN, USA
[4]Archbold Biological Field Station, 123 Main Dr., Venus, FL, USA
[5]Florida International University, Department of Biological Sciences and Institute of
Environment, Miami, FL, USA
[6]Lehigh University, Earth and Environmental Sciences Department, Bethlehem, PA, USA
[7]Miami University, Global Change Limnology Laboratory, Department of Biology, Oxford, OH,
USA
*Corresponding author: dempsey007@gannon.edu
ORCID IDs:
Dempsey: 0000-0003-3817-0155
Brentrup: 0000-0002-4818-7762
Williamson: 0000-0001-7350-1912
Gaiser: 0000-0003-2065-4821
**Keywords:** 4-6
Photodegradation, biodegradation, dissolved organic carbon, $CO_2$




**Abstract**
Outgassing of carbon dioxide ($CO_2$) from freshwater ecosystems comprises 12-25% of the total
carbon flux from soils and bedrock. This $CO_2$ is largely derived from both biodegradation and
photodegradation of terrestrial dissolved organic carbon (DOC) entering lakes from wetlands and
soils in the watersheds of lakes. In spite of the significance of these two processes in regulating
rates of $CO_2$ outgassing, their relative importance remains poorly understood in lake ecosystems.
In this study, we used groundwater from the watersheds of one subtropical and three temperate
lakes of differing trophic status to simulate the effects of increases in terrestrial DOC from storm
events. We assessed the relative importance of biodegradation and photodegradation in oxidizing
DOC to $CO_2$. We measured changes in DOC concentration, the optical characteristics of the
DOC ($SUVA_{320}$ and $S_r$), dissolved oxygen, and dissolved inorganic carbon (DIC) in short-term
experiments from May-August, 2016. In all lakes, photodegradation led to larger changes in
DOC and DIC concentrations and optical characteristics than biodegradation. A descriptive
discriminant analysis showed that in brown-water lakes, photodegradation led to the largest
declines in DOC concentration. In these brown-water systems, ~30% of the DOC was processed
by sunlight and ~2% was photo mineralized. In addition to documenting the importance of
photodegradation in lakes, these results also highlight how lakes in the future may respond to
changes in DOC inputs.




## Introduction

Lakes are closely linked to their surrounding terrestrial ecosystems. As the lowest point in the landscape, they receive a significant influx of terrestrially-derived dissolved organic carbon (DOC) and nutrients (Williamson et al., 2009; Wilkinson et al., 2013). Climate and land use changes are altering the link between lakes and their surrounding landscapes by strengthening the flow of material during extreme rain events and large wildfires, or weakening it during extended periods of drought (Strock et al., 2016; Williamson et al., 2016). Long-term changes in DOC concentrations are variable and appear to be regionally controlled. In northeastern North American and western European lakes, there has been as much as a doubling of DOC concentrations due to recovery from anthropogenic acidification and climate change (Monteith et al., 2007; Williamson et al., 2015; de Wit et al., 2016). However, DOC concentrations in Greenland lakes (Saros et al., 2015) and the Mississippi River (Duan et al., 2017) have been decreasing. A long-term study of the Florida Everglades showed that some study sites were decreasing in DOC concentration, but the majority of sites were not changing (Julian et al., 2017). As DOC inputs into aquatic ecosystems have increased, stabilized, or decreased, long-term studies have focused on understanding the mechanisms behind the change, but less research has addressed the fate of DOC once it enters a lake.

By attenuating light in the water column and also providing a source of energy, DOC serves an important role in lakes by regulating the balance between photosynthesis and respiration (Williamson et al., 1999), and thus the flux of $CO_2$ to the atmosphere (Cole et al., 1994). Previous studies indicated that most lakes are net heterotrophic, where the breakdown of organic carbon exceeds production (Kling et al., 1991; Cole et al., 1994). Estimates suggest that lakes respire about half of the annual 2 gigaton flux of carbon to the oceans each year as $CO_2$



(Cole et al., 1994; Tranvik et al., 2009; Tranvik, 2014). The traditional paradigm has been that
the dominant mechanism causing the release of excess $CO_2$ from lakes is bacterial respiration of
DOC (biodegradation), with photomineralization accounting for only 10% of bacterial rates
(Granéli et al., 1996; del Giorgio et al., 1997; Jonsson et al., 2001). However, research on over
200 Arctic lakes, rivers, and streams revealed that sunlight dominated the processing of DOC,
and photomineralization rates were on average 5x greater than dark bacterial respiration rates
(Cory et al., 2014). In addition, the source of inland water $CO_2$ remains uncertain (Raymond et
al., 2013; Lapierre et al., 2013; Weyhenmeyer et al., 2015) and predicting DOC reactivity has
been challenging (Evans et al., 2017). Quantifying the dominant degradation pathways for
terrestrial DOC from a range of lakes will improve estimates of carbon fluxes, particularly
mineralization rates that currently have a high degree of uncertainty (Hanson et al., 2014).
Many past studies have focused on testing the effects of photodegradation and
biodegradation on DOC quantity individually, but they have not simultaneously evaluated how
these two processes alter the absorbance characteristics of DOC, hereafter referred to as colored
dissolved organic matter (CDOM) (Granéli et al., 1996; Koehler et al., 2014; Vachon et al.,
2016a). The effects of sunlight on DOC are not isolated to only increasing mineralization rates;
photodegradation can also decrease the color and molecular weight of DOC, which can increase
light availability and the subsequent bacterial respiration of DOC (Bertilsson and Tranvik, 2000;
Amado et al., 2003; Chen and Jaffé, 2016). Cory et al. (2014) found the dominant degradation
process for Arctic lakes to be partial photodegradation, suggesting that in lakes, sunlight-driven
changes in CDOM without undergoing complete mineralization may dominate DOC processing.
Since light attenuation varies so strongly among lakes of differing trophic status, testing
the relative importance of DOC processing via photodegradation or biodegradation with





mechanistic experiments is needed. Previous research on DOC degradation has primarily
occurred in high DOC lakes, but in clear-water lakes, 1% of surface UV-A and
photosynthetically active radiation (PAR), which are the primary wavelengths active in
photodegradation (Osburn et al., 2001), may reach up to 5-7 m for UV-A and 12-14 m for PAR
in oligotrophic lakes and depths of  >45-50 m in some of the clearest lakes in the world, such as
Lake Tahoe (Rose et al., 2009a; Rose et al., 2009b). Geographic location and time of year
influence the amount of solar radiation lakes receive. In the subtropics, PAR and UV light have
high intensity across the spectrum year-around, whereas in temperate regions those wavelengths
are strongest during the summer months.

Watershed land use and lake trophic status have also been shown to influence DOC

composition and reactivity (Lu et al., 2013; Hosen et al., 2014; Larson et al., 2014; Evans et al.,
2017). DOC from forested systems was more reactive when compared to disturbed environments
and had different optical properties (Lu et al., 2012; Williams et al., 2015; Evans et al., 2017).
Studies examining how terrestrial DOC inputs are processed in lakes are needed, especially with
the increasing frequency of extreme rain events (Rahmstorf and Coumou, 2011; Westra et al.,
2014; Fischer and Knutti, 2015). Future climate change projections suggest that for northern
ecosystems a 10% increase in precipitation could lead to a 30% increase in the mobilization of
soil organic matter (de Wit et al., 2016). Extreme rain events deliver fresh DOC not exposed to
prior sunlight into lakes, which can lead to significant reductions in light availability, as well as
increases in thermal stability and lake heterotrophy (Jennings et al., 2012; Klug et al., 2012; de
Eyto et al., 2016; Zwart et al., 2016). As DOC concentrations change globally, understanding the
processes that determine the fate of DOC will help predict the systems most likely to release
more $CO_2$.



Here our aim was to 1) determine the relative importance of photodegradation and
biodegradation for altering terrestrial DOC quantity and CDOM from lakes of varying trophic
status, 2) quantify the percentage of the initial DOC pool that was photomineralized, partially-
photodegraded, biodegraded or remained unprocessed, and 3) compare the effects of
photodegradation on DOC quantity and CDOM across four lakes to understand differences in
how terrestrial DOC from the watersheds of different lake types responds to photodegradation.
Since lakes are closely linked to their surrounding landscape (i.e. soils and vegetation), we
collected terrestrial DOC from the watershed of three temperate lakes and one subtropical lake,
all varying in trophic status. This soil organic matter represents the current and future inputs of
organic material. We studied changes in the concentration of DOC, dissolved inorganic carbon
(DIC), and dissolved oxygen (DO) and measured changes in CDOM. We hypothesized that
photodegradation would be more important than biodegradation in all lakes, but the strongest
responses to sunlight would be observed in the brown-water lakes.

**1. Methods**
**1.1** *Study Sites and Samplers*
Groundwater samples were collected from the watersheds immediately adjacent to four
lakes used in this study (Table 1). All of the lakes are small, with a surface area $\leq 0.48$ km$^2$ and a
maximum depth ranging from 12.5 m in Lake Waynewood to 24 m in Lake Giles. The three
temperate lakes (Giles – oligotrophic; Lacawac – brown-water; Waynewood – eutrophic) are in
close proximity, located on the Pocono Plateau in northeastern Pennsylvania. Lake Annie
(brown-water) is a subtropical, sinkhole lake located on the Lake Wales Ridge in south-central
Florida. These lakes were selected because of their variability in the dominant vegetation types



in their watersheds that lead to differences in DOC concentration and quality (Table 1). Annie,
Giles, and Lacawac are all seepage lakes within protected watersheds, and there have been no
significant changes in land use or land cover over the past thirty years. The watersheds of Giles
and Lacawac have > 90% cover of mixed and northern hardwood-conifer forests, with oak trees
dominating the watershed at Giles, while hemlocks represent the highest proportion of
Lacawac's watershed (Moeller et al., 1995). Annie is surrounded by well-drained sandy soils and
the major vegetation types include a mixed-scrub community, pinelands, and oak forests (Gaiser,
2009). Both Annie and Lacawac are brown-water lakes with moderate DOC concentrations and
lower transparency (Table 1). A higher percentage of wetlands (7% for Annie and 25% for
Lacawac) in their watersheds likely contribute to their darker color compared to the other lakes
(Moeller et al., 1995; H. Swain *unpublished data)*. Waynewood is the most eutrophic lake and
has the largest watershed with runoff from dairy farms upstream that feeds into the lake through
an inlet stream. The forest surrounding Waynewood is similarly dominated by oak and hemlock
trees, but there is overall less total forest cover in the watershed than Lacawac and Giles, and
there are more homes adjacent to the lake (Moeller et al., 1995).



**Table 1**. Summary characteristics of the four study lakes in May-August 2013–2016 (mean ± SD). Abbreviations: Chl-*a* (chlorophyll-a), DOC (dissolved organic carbon), GW DOC (initial groundwater DOC), PAR (photosynthetically active radiation, 400-700 nm), UV-A (ultraviolet A radiation, 380 nm), UV-B (ultraviolet B radiation, 320 nm), RT (residence time).

| Lake | Lat. (°) | Long. (°) | Lake area (km²) | Max. depth (m) | Chl-*a* (µg L⁻¹) ± (SD) | Lake DOC (mg L⁻¹) ± (SD) | GW DOC (mg L⁻¹) ± (SD) | pH ± (SD) | 1% UV-B depth (m) ± (SD) | 1% UV-A depth (m) ± (SD) | 1% PAR depth (m) ± (SD) | RT (yr) |
|---|---|---|---|---|---|---|---|---|---|---|---|---|
| Lacawac | 41° 22′ N | 75° 17′ W | 0.21 | 13 | 1.9 (1.4) | 5.2 (0.8) | 59.4 (6.1) | 6.6[+] | 0.4 (0.1) | 0.9 (0.2) | 5.7 (0.6) | 3.3 |
| Annie | 27° 12′ N | 81° 20′ W | 0.36 | 20.7 | 4.0 (1.5) | 9.4 (2.5) | 20.7 (0.5) | 5.5 (0.3) | 0.5* | 1.3* | 4.5 (1.6) | 2 |
| Wayne-wood | 41° 23′ N | 75° 21′ W | 0.28 | 12.5 | 5.3 (3.7) | 6.4 (1.0) | 7.6 (0.3) | 7.5[+] | 0.3 (0.1) | 0.7 (0.2) | 4.3 (0.9) | 0.42 |
| Giles | 41° 22′ N | 75° 5′ W | 0.48 | 24 | 1.1 (0.7) | 2.3 (0.3) | 6.0 (0.6) | 6.2[+] (0.3) | 2.0 (0.5) | 4.7 (1.2) | 14.4 (2.1) | 5.6 |

*Indicates estimates from a single profile in March 2012. [+]pH data in Lacawac and Waynewood are from 2015 only and from 2015-2016 in Giles.

Samplers were used to collect groundwater as a proxy for terrestrial DOC runoff entering the lakes. The samplers were installed in close proximity to the Pocono lakes near small inlet streams in sandy or bog areas on 6 July 2015 (~1 year prior to experiments). The groundwater sampler consisted of 1m sections of 7.6cm diameter PVC pipe installed to a depth of 60-81cm below ground. 0.5cm holes were drilled in the sides with a fine mesh covering the holes to let shallow groundwater in but exclude large particulates. At Lake Annie, a groundwater sampler was installed on 17 March 2016 on the south side of the lake near a small, intermittent inlet stream. The groundwater sampler near Lake Annie was a 3m section of PVC pipe installed slightly deeper to 2m below ground to allow continuous access to groundwater during the dry season.



On 7 May 2016, 10 L of water was collected using a peristaltic pump from the

groundwater samplers at all of the Pocono lakes in acid-washed 18 L bottles. Groundwater
samples from Annie were collected from the sampler monthly (25 April, 31 May, 27 June, and 1
Aug 2016) prior to starting the experiments and shipped overnight on ice to Pennsylvania. All
groundwater samples were kept cold (4 °C) and dark until filtered to avoid sunlight exposure
prior to the start of the experiments. Samples for the May experiments were filtered on May 8,
2016 through a 0.7 µm Whatman GF/F filter. The remaining 8 L of groundwater for the June,
July, and August experiments for each Pocono lake were filtered in a similar manner over the
next 14 days. Samples were kept cold and dark until the experiments started. Samples for June,
July, and August were re-filtered with a 0.7 µm Whatman GF/F filter prior to the start of those
experiments. The initial DOC concentration of the groundwater for each lake varied at the start
of each experiment, but it was always higher than the in-lake DOC concentration (Table 1).

**1.2 *Sampling Design and Variables Analyzed***

To determine the relative importance of photodegradation and biodegradation for

processing DOC, we designed three treatments in a manner similar to Cory et al., (2014): 1)
photodegradation only, 2) biodegradation only, and 3) control. From each treatment, five
different variables were measured including DOC concentration, DIC concentration, DO
concentration, $SUVA_{320}$, and $S_r$. The different variables measured in each treatment required the
use of different containers for the sample water. Samples for DOC analysis (concentration and
CDOM) were deployed in acid-washed, muffled 35 mL quartz tubes sealed with silicone
stoppers. The quartz tubes had an average transmittance of 96% of solar UV-A and 87% of solar
UV-B, which allowed for an accurate representation of *in-situ* solar radiation levels (SFig. 1,



Morris and Hargreaves, 1997). However, the quartz tubes were not gas tight, so samples for
dissolved inorganic carbon (DIC) and dissolved oxygen (DO) analysis were deployed in gas tight
borosilicate vials (Labco, Ceredigion, UK). The borosilicate vials had a volume of 12 mL but
were filled to 10 mL due to safety concerns with mercury chloride (see below). A clean 10 mL
pipette was used to carefully transfer water into the borosilicate vials. Borosilicate glass has a
sharp cut-off at 320 nm and transmits <5% UV-B, but it transmits an average of 63% of UV-A
radiation and 90% of PAR (SFig. 1, Reche et al., 1999).

Water samples for all of the treatments were initially filtered through ashed 0.7 μm

Whatman GF/F filters one day prior to the start of each monthly experiment. For the
photodegradation and control treatments detailed below, samples for DO and DIC analysis were
treated with 0.35 mL of 1% mercury chloride ($HgCl_2$) to kill the microbial community. Samples
for DOC concentration and CDOM analysis ($SUVA_{320}$ and $S_r$) for the same treatments were
sterile filtered with a 0.2 μm membrane filter (Sterivex MilliporeSigma, Burlington, MA USA)
pre-rinsed with 100 mL of DI water and 50 mL of sample water instead of using $HgCl_2$ because
adding $HgCl_2$ altered the optical scans. Sterile filtering has previously been shown to remove the
majority of microbes present, and water samples remained sterile for one week following this
procedure (Moran et al., 2000; Fasching and Battin, 2011). For the biodegradation treatment,
water samples were inoculated with 100 μL of unfiltered groundwater that was collected 1 day
prior to the start of each monthly experiment. By adding a fresh inoculum of groundwater each
month, we aimed to re-stimulate the microbial community and assess the short-term response of
biodegradation. Treatments were deployed in triplicate for each lake (i.e. 3 DOC quartz tubes, 3
DO borosilicate vials, and 3 DIC borosilicate vials for each treatment). Here, we included a
summary of the three experimental treatments that were designed as follows:





a) *Photodegradation Only*: Water for DOC concentration and CDOM analysis (SUVA$_{320}$
and S$_r$) was sterile filtered and stored in quartz tubes (n = 3 replicates). Water for DIC
and DO analysis was treated with 1% HgCl$_2$ and stored in borosilicate vials (n = 6
replicates; 3 replicates for DIC and 3 replicates for DO analysis).
b) *Biodegradation Only*: Water for all analyses was inoculated with 100 μL of unfiltered
groundwater. Water samples for DOC concentration and CDOM analysis were stored in
quartz tubes (n = 3 replicates). Water samples for DIC and DO analysis were stored in
borosilicate vials (n = 6 replicates; 3 replicates for DIC and 3 replicates for DO analysis).
Both the quartz tubes and borosilicate vials were wrapped with multiple layers of
aluminum foil to eliminate light exposure.
c) *Control*: Water for DOC concentration and CDOM analysis was sterile filtered and
stored in quartz tubes (n = 3 replicates). Water for DIC and DO analysis was treated with
1% HgCl$_2$ and stored in borosilicate vials (n = 6 replicates; 3 replicates for DIC and 3
replicates for DO analysis). All samples were wrapped in aluminum foil (dark).

The experimental treatments for each lake were deployed for seven days at the surface of
Lake Lacawac in May, June, July, and August of 2016 (for exact sampling dates see SI, Table 1).
Samples were kept at the lake surface using floating racks, and samples from each lake were
randomly distributed across the racks. The deployment design ensured that samples stayed at the
surface and dipped no deeper than 2 cm in the water column. After the one-week exposure, racks
were collected from the surface of Lake Lacawac and samples were immediately transferred into
coolers and returned to the lab. We assessed the response of terrestrially derived DOC to
photodegradation and biodegradation by measuring changes in the concentrations of DOC, DIC,



and DO, and the absorbance properties ($SUVA_{320}$ and $S_r$) of the CDOM. All samples were
analyzed within 72 hours of collection.

Dissolved organic carbon concentrations were analyzed using a Shimadzu TOC-V$_{CPH}$

Total Organic Analyzer with an ASI-V auto sampler. External acidification was used for each
sample and triplicate measurements were performed following the methods of Sharp (1993).
Dissolved inorganic carbon concentrations (as $CO_2$) were measured with a Shimadzu GC-8A
Gas Chromatograph using helium as the carrier gas. Samples were acidified using 0.1 N $H_2SO_4$
and then stripped with nitrogen gas prior to injection. Dissolved oxygen was measured using a
modified Winkler titration (Parson et al., 1984). Samples for gas measurements (DO and DIC)
were kept in a 21°C water bath for 30 minutes prior to analysis. These samples were well mixed
just prior to analysis. The absorbance properties of CDOM were analyzed using a Shimadzu UV
1800 scanning spectrophotometer at 25°C. Raw absorbance scans were generated from 800 to
200 nm using a 1 cm cuvette and were blank corrected with ultra-pure DI water. From the
absorbance scans, the spectral slope ratio ($S_r$) was calculated following Helms et al., (2008). The
DOC specific ultraviolet absorbance at 320 nm ($SUVA_{320}$) was calculated following methods in
Williamson et al., (2014). $S_r$ can be used as a proxy for the molecular weight of the DOC, while
$SUVA_{320}$ can be used as a proxy for DOC color and aromatic carbon content (Helms et al., 2008,
Williamson et al., 2014).

Due to differences between the borosilicate vials and quartz tubes, the DIC and DO

samples were spectrally corrected for the amount of light they received (SI, SFig. 1). Total
cumulative energy exposure over the monthly incubations was calculated from a BSI Model
GUV-521 (Biospherical Instruments, San Diego, CA) radiometer with cosine irradiance sensors
that have a nominal bandwidth of 8 nm for 305 nm, 320 nm, 340 nm, 380 nm, and 400-700 nm



(PAR). Daily irradiance for UV-B, UV-A, and PAR were calculated using 15-minute averages of
1-second readings from a GUV radiometer located near Lake Lacawac over the 7-day
experiments. The area under the curve was calculated by multiplying the measurement frequency
(900 sec) by the average of two adjacent time step readings. These values were then summed
over the exposure period to calculate the total cumulative energy exposure for each sample.
Readings from a profiling BIC sensor (Biospherical Instruments, San Diego, CA) were then used
to calculate the percent of the deck cell at the surface rack incubation depth (0.02 m) in Lake
Lacawac.

**1.3** *Explanation of Calculations and Statistical Analysis*

To determine the fate of terrestrial DOC in the four lakes, we used the measured changes

(i.e. final – control) in DOC and DIC concentrations to identify four pools of DOC:
photomineralized, partially photodegraded, biodegraded, and unprocessed. Each pool was
converted to a carbon basis, and we assumed a conversion of 0.5 moles $CO_2$ for each mole of
DOC consumed (Cory et al., 2014). The amount of carbon photomineralized (converted to $CO_2$)
was calculated as the concentration of DIC produced [DIC*2] by sunlight (i.e. carbon that was
completely oxidized by sunlight). The amount of carbon partially photodegraded represents the
remainder of the carbon pool that was processed by sunlight (but not completely oxidized to
$CO_2$) and was calculated as the total DOC processed by sunlight minus the amount
photomineralized [Total Photodegraded – Photomineralized]. The amount of carbon biodegraded
was calculated as the concentration of DOC lost in the biodegradation treatments. The
unprocessed carbon was calculated as the fraction of the carbon pool that was not processed by
either sunlight or microbes [Control DOC – Photomineralized – Partially Photodegraded –



Biodegraded]. Each process was determined for each lake and each month. Here we report the
average response across all four months for each DOC pool.
While we carried out monthly experiments (May-August), here we report the average
response across the open-water season (i.e. all four months) to provide a more complete picture
of DOC processing. The downside of this approach is that it potentially increases variation in
variables associated with DOC processing, since such processing may vary across the season.
However, there was not a strong seasonal response to photodegradation or biodegradation in all
of our study variables (SI Fig. 3). Furthermore, the majority of the terrestrial DOC was collected
on a single date and time (except for Lake Annie).
Final treatments were compared relative to the dark and killed (1% $HgCl_2$) control
treatments, as those samples were deployed at the surface of the lake with the photodegradation
and biodegradation treatments. We used a t-test to determine whether the photodegradation
samples for all of the variables were significantly different from the biodegradation samples (n =
12 for each treatment) in each lake. Photodegradation and biodegradation samples were analyzed
separately using a one-way ANOVA to assess differences between lakes. A post-hoc Tukey's
multiple comparison test (Sigma Plot 14.0) was used to determine if there were significant
differences in the response variables between the lakes to the photodegradation and
biodegradation treatments. A descriptive discriminant analysis (DDA) was used to classify the
four lakes based on changes in DOC, DIC, DO, $SUVA_{320}$, and $S_r$ measurements due to
photodegradation. Since these five measures are likely to be highly correlated with one another,
DDA is a good choice since it considers these relationships simultaneously in the analysis
(Sherry 2006). In this case, DDA, works by producing linear combinations of the five measured
variables (DOC, DIC, DO, $SUVA_{320}$, and $S_r$). The first linear combination provides the best





separation of the four lakes, followed by subsequent linear combinations for axes that are
orthogonal (Sherry, 2006). Linear combinations are weighted more heavily by variables that are
better able to discriminate between the lakes. In the figures and tables below, we report these
data as either average measured changes (i.e. concentrations) or average percent changes and
have indicated where appropriate. Data for this experiment were analyzed in either Sigma Plot
14.0 (Fig. 1, Table 2) or Systat version 10.2 (Fig. 4).

**2. Results**

Throughout the results and discussion, the use of the lake names is to present the data in a

meaningful manner, but it is important to recognize that the actual water samples originated from
groundwater samples adjacent to each lake.

**2.1 *Photodegradation and biodegradation responses in each lake***

Photodegradation altered DOC quantity and CDOM significantly more than

biodegradation for terrestrial DOC from the watersheds of all four lakes (Table 2, Fig. 1). For the
photodegradation only treatments, exposure to sunlight resulted in significant production of DIC
and increases in $S_r$, as well as significant decreases in DO, DOC, and $SUVA_{320}$ relative to the
biodegradation treatments. The only significant effect of biodegradation on terrestrial DOC was
a reduction in DO concentrations compared to the dark control (Fig. 1c). In all other cases, the
biodegradation treatments were not significantly different than the control, and the average
percent change was close to 0.

The terrestrial DOC from the brown-water lakes (Lacawac and Annie) typically followed

similar patterns to each other, while the terrestrial DOC from the oligotrophic and eutrophic



lakes (Giles and Waynewood) responded more similarly to each other. In the brown-water lakes,
we observed a stronger response in DOC quantity (i.e. DOC, DIC, and DO), while the changes in
DOC quantity were much more muted in the oligotrophic and eutrophic lakes. The responses of
$S_r$ changes in each lake due to sunlight did not differ significantly. All four lakes showed a strong
response to changes in terrestrial CDOM (i.e. $SUVA_{320}$ and $S_r$).





**Table 2.** A summary of the mean (± SD) final concentration of DOC, DIC, DO, SUVA$_{320}$ and S$_r$
in photodegradation (Photo), biodegradation (Bio), and control experimental treatments in
groundwater samples from the watersheds of lakes Lacawac, Annie, Giles, and Waynewood. The
mean (± SD) initial concentration for each variable is also depicted. The P/B column list the
results of a t-test to determine whether photodegradation samples were significantly different
from the biodegradation samples (n = 12 for each treatment for the four months). Bolded values
indicate the Photo treatments that were statistically different from the Bio treatments (p < 0.05).

| Analysis | Treatment | Lacawac (Mean ± SD) | P/B p-value | Annie (Mean ± SD) | P/B p-value | Giles (Mean ± SD) | P/B p-value | Waynewood (Mean ± SD) | P/B p-value |
|---|---|---|---|---|---|---|---|---|---|
| DOC (μmoles L$^{-1}$) | Photo | 3600 ± 330 | **p < 0.001** | 1270 ± 211 | **p < 0.001** | 692 ± 123 | **p < 0.001** | 883 ± 73.3 | **p < 0.001** |
| | Bio | 4910 ± 674 | | 1810 ± 45.7 | | 608 ± 99.0 | | 765 ± 93.8 | |
| | Control | 5110 ± 628 | | 1820 ± 76.9 | | 630 ± 102 | | 783 ± 73.8 | |
| DIC (μmoles L$^{-1}$) | Photo | 54 ± 8.2 | **p < 0.001** | 41.9 ± 11.4 | **p < 0.001** | 20.4 ± 1.9 | **p < 0.001** | 32.2 ± 7.3 | **p = 0.02** |
| | Bio | 16.1 ± 5.0 | | 25.3 ± 7.2 | | 17.7 ± 3.0 | | 27.1 ± 8.0 | |
| | Control | 13.8 ± 4.6 | | 30.4 ± 18.2 | | 15.3 ± 2.1 | | 27.8 ± 3.5 | |
| DO (μmoles L$^{-1}$) | Photo | 278 ± 62.4 | **p < 0.001** | 419 ± 25.9 | **p < 0.001** | 536 ± 35.6 | p = 0.09 | 522 ± 49.0 | **p < 0.001** |
| | Bio | 556 ± 46.4 | | 533 ± 42.2 | | 556 ± 34.3 | | 577 ± 76.9 | |
| | Control | 660 ± 29.4 | | 656 ± 32.1 | | 688 ± 60.9 | | 702 ± 57.3 | |
| SUVA$_{320}$ (m$^{-1}$/mg L$^{-1}$) | Photo | 4.3 ± 0.4 | **p < 0.001** | 2.4 ± 0.4 | **p < 0.001** | 2.4 ± 0.2 | **p < 0.001** | 1.8 ± 0.2 | **p < 0.001** |
| | Bio | 5.3 ± 0.2 | | 3.8 ± 0.1 | | 4.8 ± 0.3 | | 3.2 ± 0.2 | |
| | Control | 5.1 ± 0.2 | | 3.8 ± 0.1 | | 4.7 ± 0.2 | | 3.2 ± 0.1 | |
| S$_r$ | Photo | 1.1 ± 0.0 | **p < 0.001** | 1.3 ± 0.1 | **p < 0.001** | 1.4 ± 0.1 | **p < 0.001** | 1.2 ± 0.1 | **p < 0.001** |
| | Bio | 0.7 ± 0.1 | | 0.8 ± 0.0 | | 0.9 ± 0.1 | | 0.8 ± 0.1 | |
| | Control | 0.7 ± 0.1 | | 0.8 ± 0.0 | | 0.9 ± 0.1 | | 0.9 ± 0.1 | |


Sunlight caused average (± SD) DOC losses relative to the control treatments of 30.5 ±
11.5% and 28.9 ± 8.3% in Lacawac and Annie, respectively (Fig. 1a). In Giles and Waynewood,
we observed an average of 9.6 ± 6.5% and 13.4 ± 6.2% increase in DOC concentration,
respectively following exposure to sunlight. When we compared lakes within each treatment,
there were no significant differences in DOC concentration due to sunlight in Giles vs.
Waynewood, whereas Annie and Lacawac were significantly different from the prior two lakes
and from each other (ANOVA: $F_{1,3}$ = 70.9, p < 0.001).
Decreases in DOC concentration due to photodegradation could lead to mineralization
(i.e. DIC production; Fig. 1b) and therefore oxidation (i.e. DO consumption; Fig. 1c). We



observed the production of DIC due to sunlight in all of our lakes (Fig. 1b). In Lacawac and
Annie, the average ($\pm$ SD) percent increases in DIC relative to the control treatments were $350 \pm$
160% and $96.0 \pm 79.0\%$, respectively. The average percent increases relative to controls in Giles
and Waynewood were $40.7 \pm 19.4\%$ and $23.2 \pm 12.7\%$ respectively. The DIC percent change
was similar between Giles and Waynewood, and both were statistically different from Annie and
Lacawac. The percent DIC change in Lacawac was significantly higher than Annie (ANOVA:
$F_{1,3} = 36.4$, $p < 0.001$).

In all lakes, both photodegradation and biodegradation led to decreases in DO

concentrations (Fig. 1c). Average DO losses due to biodegradation for all four lakes ranged from
15 to 18%. DO losses due to photodegradation were more variable. The average DO loss from
sunlight in Lacawac and Annie was $58.2 \pm 7.8\%$ and $35.9 \pm 5.4\%$, respectively. In Giles and
Waynewood, we observed average DO losses of $21.6 \pm 7.9\%$ and $25.6 \pm 4.7\%$ respectively.
While the largest losses of DO due to sunlight were observed in Annie and Lacawac, there was
no significant difference between Annie and Waynewood. Giles and Lacawac were significantly
different from the other two lakes and from each other (ANOVA: $F_{1,3} = 73.9$, $p < 0.001$).

Changes in CDOM due to biodegradation were minimal in all of the lakes (Fig. 1d & 1e).

In contrast, photodegradation caused significant changes in all of the lakes, but the magnitude of
the change varied by lake. $SUVA_{320}$ decreased in all lakes due to sunlight, but the largest changes
were observed in the oligotrophic and eutrophic lakes (Fig. 1d). Average $SUVA_{320}$ values
decreased between 16.8% in Lacawac and 48.9% in Giles. The response in Annie and
Waynewood were similar, whereas Lacawac and Giles were significantly different from the prior
two lakes and each other (ANOVA: $F_{1,3} = 39.7$, $p < 0.001$). In all lakes, $S_r$ increased due to
sunlight (Fig. 1e). Average percent increases for the lakes ranged from 46.4% in Waynewood to



400 65.1% in Lacawac. For $S_r$, the response between Lacawac and Waynewood were significantly

401 different, but those lakes were no different compared to the remaining lakes (ANOVA: $F_{1,3} = 3.1$,

402 p = 0.04).


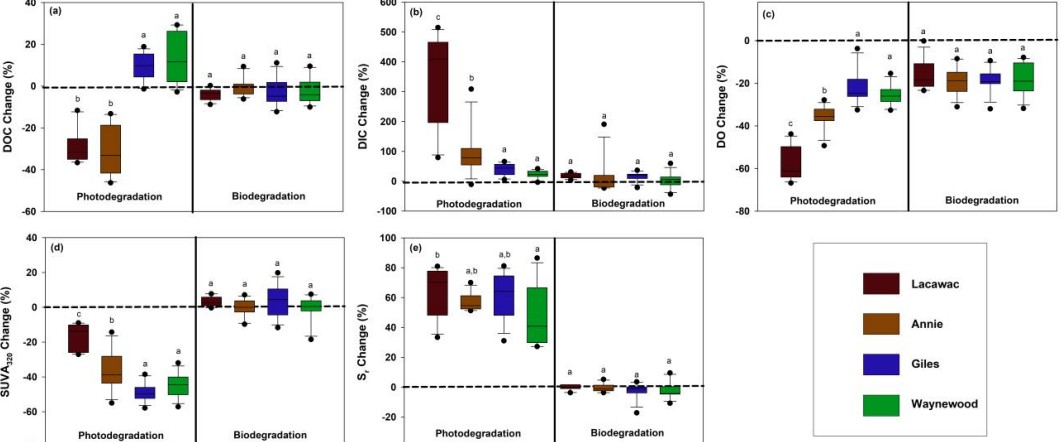


405 **Figure 1**. The monthly average percent change from the dark and killed control treatments
406 (dashed line) in each lake for photodegradation (left) and biodegradation (right) for (a) DOC, (b)
407 DIC, (c) DO, (d) $SUVA_{320}$, and (e) $S_r$. Statistical differences (p < 0.05) between lakes are
408 indicated by different letters above each boxplot. For each boxplot n =12 replicates.

410 **2.2 Fate of DOC**

411  Of the four pools of carbon we identified in the groundwater samples entering our study

412 lakes, we found the average amount of carbon processed by sunlight ranged from 1.2% to ~30%

413 (Fig. 2). Carbon in Giles and Waynewood (< 2%) showed little response to sunlight, whereas the

414 response in Annie and Lacawac (~30%) was much higher over the 7-day experiments. The

415 dominant pathway through which sunlight interacted with DOC was through partial

416 photodegradation in these latter two lakes. About 2% of the carbon pool was photomineralized in

417 the brown water lakes. The amount of carbon processed via biodegradation was minimal in all


lakes (ranging from 0.2–4%). The fraction of the unprocessed carbon pool ranged from a low of
66% for Lacawac to a high of 97% for Waynewood.  An average of 3.4 to 34% of the carbon
pool was processed in one week.

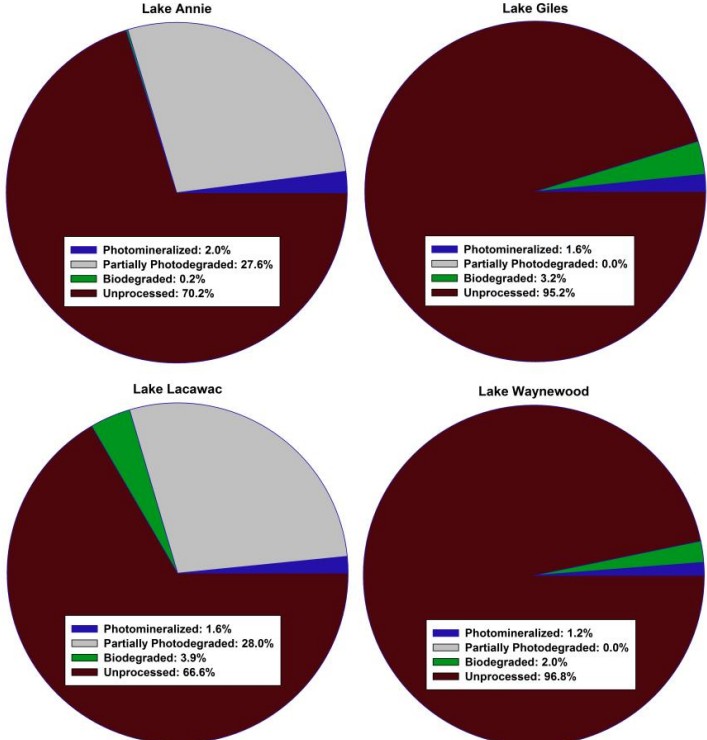


**Figure 2.** A summary of the average fate of carbon in the groundwater samples from our study
lakes (see methods section for explanation of calculations). All terms were converted to a carbon
basis. Photomineralized describes the amount of carbon completely mineralized to $CO_2$ by
sunlight. Partially photodegraded describes the amount of carbon processed by sunlight minus
the amount photomineralized. Biodegraded describes the amount of carbon lost through
biodegradation. Unprocessed carbon describes the remaining carbon that was not processed by
photodegradation or biodegradation.


**2.3 *DOC response by lake trophic status***
For the descriptive discriminant analysis (DDA) to classify the lakes, we found that the
five metrics were strongly correlated with one another (Table 3). In general, the changes in DOC,



DIC, and DO were more strongly correlated with one another than with $SUVA_{320}$ and $S_r$ and vice
versa (Table 3). We will refer to the changes in DOC, DIC, and DO as "DOC quantity" and the
changes in $SUVA_{320}$ and $S_r$ as "CDOM" for brevity.

**Table 3**. Pearson correlations between the measured changes in the five metrics: DOC, DIC, DO,
$SUVA_{320}$, and $S_r$.

|  | DOC | DIC | DO | $SUVA_{320}$ |
|---|---|---|---|---|
| DIC | -0.934 |  |  |  |
| DO | 0.869 | -0.837 |  |  |
| $SUVA_{320}$ | -0.705 | -0.671 | -0.666 |  |
| $S_r$ | -0.027 | 0.021 | 0.163 | -0.319 |

DDA produced three functions (axes) with canonical correlations of 0.961, 0.753, and

0.181 (Fig. 3). Collectively, the entire model was significant (Wilks' $\lambda = 0.032$; $F_{15,\ 108} = 17.79$; p
< 0.001). Effect size was calculated following Sherry and Henson (2010) as $1 - $ Wilks' $\lambda$, and
therefore the overall model explains 96.8% of the variation among lakes. Functions 1 through 3
and 2 through 3 were significant (p < 0.001 for both). Function 3 was not significant (p = 0.710)
and therefore is not discussed further. Functions 1 through 3 collectively explain 92.4% of the
shared variance while functions 2 through 3 collectively explain 56.7% of the shared variance.

Function 1 represents a new variate that is a linear combination of the changes in the five

variables that best discriminates the lakes from one another. This new variate is composed
mainly of DOC, with a function coefficient of 0.465 and a structure coefficient of 0.821 (Table
4). Of note are also DIC, DO, and $SUVA_{320}$ that had smaller function coefficients (< 0.45), but
had large structure coefficients (> 0.45). This result suggests that Function 1 is mainly related to
DOC quantity. Function 2, also a new variate that is a linear combination of the five measured
changes, is composed mainly of $SUVA_{320}$ (function coefficient = 0.985 and structure coefficient



= 0.719; Table 4). Function 2 is orthogonal to Function 1 and together they discriminate the four
lakes (Fig. 3).
**Table 4**. The solution for changes in measured independent variables that predict the dependent
variable, lake. Structure coefficients ($r_s$) and communality coefficients greater than $|0.45|$ are in
bold. Coeff = standardized canonical function coefficient; $r_s$ = structure coefficient; $r_s^2$ = squared
structure coefficient.

|  | Function 1 | | | Function 2 | | |
|---|---|---|---|---|---|---|
| Variable | Coeff. | $r_s$ | $r_s^2(\%)$ | Coeff. | $r_s$ | $r_s^2(\%)$ |
| DOC | 0.465 | **0.821** | 67.40 | 0.639 | 0.278 | 40.83 |
| DIC | -0.337 | **-0.703** | 11.36 | -0.059 | -0.216 | 0.35 |
| DO | 0.440 | **0.679** | 19.36 | -0.124 | 0.009 | 1.54 |
| SUVA$_{320}$ | -0.139 | **-0.473** | 1.93 | 0.985 | **0.719** | 97.02 |
| S$_r$ | 0.244 | 0.068 | 5.95 | -0.238 | -0.434 | 5.66 |


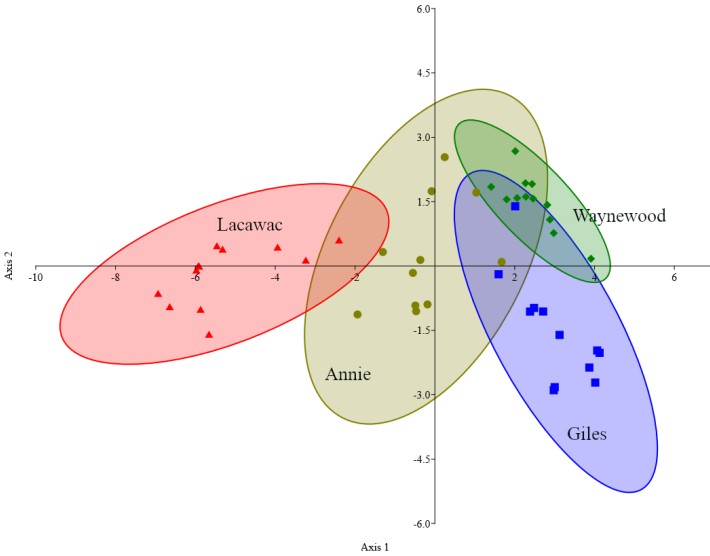



**Figure 3.** Canonical plot scores and 95% confidence ellipses from descriptive discriminant
analysis of the measured changes (i.e. treatment minus control) in the five variables (DOC, DIC,
DO, SUVA$_{320}$, and S$_r$) and four lakes: Annie (olive circles), Giles (blue squares), Lacawac (red
triangles), and Waynewood (green diamonds). Only photodegradation samples were included in
this analysis.



DDA correctly classified 89.4% of the samples to their collection site (Fig. 3). One
sample from Annie was incorrectly assigned to Waynewood, two samples from Giles were
incorrectly assigned to Waynewood, and two samples from Lacawac were incorrectly assigned
to Annie. All of the Waynewood samples were correctly classified.

**3. Discussion**
**3.1** *Comparing the relative importance of photodegradation and biodegradation*
Despite a large number of studies examining the effects of either photodegradation or
biodegradation on DOC processing, very few have conducted simultaneous *in-situ* experiments
of the relative importance of both processes for transforming DOC from the watersheds of a
range of different lakes. Our results indicate that sunlight was the primary process in the surface
waters responsible for degrading terrestrial DOC from the watershed of all four lakes.
Biodegradation played a minimal role in changing the DOC quantity and CDOM. We observed
decreases in DOC, DO, and $SUVA_{320}$ due to sunlight and saw increases in DIC and $S_r$. The loss
of DOC, as well as a shift to more photobleached, and lower molecular weight organic material
is consistent with prior studies on these lakes that evaluated just the effects of sunlight (Morris
and Hargreaves, 1997). Exceptions to DOC loss due to photodegradation occurred in Giles and
Waynewood. In these lakes, we observed an increase in average DOC concentrations. In Giles,
there was significant production of DOC in June and July. In Waynewood, significant production
occurred in May and July. We speculate that this production may be due to the lysing of any
microbes remaining in solution. Increases may also be attributed to interactions with iron. We
have no measurable evidence, but a number of samples from Giles and Waynewood contained a



red precipitate at the conclusion of the one-week experiments. Previously iron-bound DOC could
have been released back into the water.

Dissolved oxygen was the lone variable where biodegradation led to decreases in DO

relative to the controls, but the differences between lakes were not significant. We attributed the
changes in DO to the "sloppy feeding" of bacteria, where they produce DOC through exudates
and then assimilate it (Evans et al., 2017). The above results are similar to observations in Arctic
and tropical waters in that photodegradation was more important than biodegradation on short
time scales (Cory et al., 2014; Chen and Jaffé, 2014; Amado et al., 2003). Interestingly, we
found that terrestrial DOC from the watersheds of lakes of different trophic status was processed
differently, resulting in DIC production and DOC degradation for the brown-water lakes
(Lacawac and Annie), but greater changes in $SUVA_{320}$ for the oligotrophic and eutrophic lakes
(Giles and Waynewood). This highlights the need to account for lake trophic status in predicting
DOC processing and $CO_2$ emissions from lakes.

**3.2 *Dominant degradation process***

Based on our study design we were able to identify four pools of carbon:

photomineralized, partially photodegraded, biodegraded, and unprocessed. The dominant
degradation pathway across all lakes was partial photodegradation (i.e. loss of DOC, but no
mineralization), although the size of each carbon pool varied by lake. In the brown-water lakes,
~28% of the total carbon pool was partially photodegraded and ~2% was photomineralized. In
the oligotrophic and eutrophic lakes ~1.4% of the carbon was photodegraded and none of the
carbon was photomineralized.



Observations in Toolik Lake showed 70% of the total carbon pool being processed by
sunlight during the open water period (~3 months) (Cory et al., 2014). Other estimates have
found that photomineralization of DOC accounts for only 8-14% of total water column $CO_2$
production (Granéli et al., 1996; Jonsson et al., 2001; Koehler et al., 2014; Vachon et al., 2016b).
We observed 30% of the carbon pool being processed by sunlight within one week in our lakes
and this was restricted to the brown-water lakes. Similar to Toolik Lake, the dominant
degradation process was partial photodegradation. Partial photodegradation can alter CDOM and
stimulate subsequent bacterial respiration. Degradation of CDOM can have important effects for
downstream ecosystems if it can be further processed and released as $CO_2$ or instead is buried or
exported downstream (Weyhenmeyer et al., 2012; Catalan et al., 2016; Chen and Jaffe, 2014;
Biddanda and Cotner, 2003). It is thus important to include all sunlight-driven degradation
processes to fully account for its relative importance.
Differences between the responses observed in the Arctic and our temperate/subtropical
lakes are most likely explained by the initial concentration and quality of terrestrially derived
DOC. In the Arctic, glacial meltwater can be highly photolabile and dominated by seasonal
inputs of DOC from shallow or deep soils (Cory et al., 2014; Spencer et al., 2014; and Kaiser et
al., 2017). In temperate regions, DOC tends to contain more humic and fulvic acids derived from
soils, which may be less photolabile than Arctic DOC. Additionally, we did not integrate our
results over the entire water column because the samples were analyzed on the surface of a single
lake. Over the entire water column, photodegradation could have processed additional carbon. In
clear-water lakes, DOC may be photodegraded down to the 1% UV-A attenuation depth (Osburn
et al., 2001), which ranged from 0.7-4.7 m in our study lakes (Table 1).





### 3.3 *Response of lakes to photodegradation*


With an increase in extreme precipitation events, terrestrial DOC inputs are likely to
increase in many aquatic ecosystems (Rahmstorf and Coumou, 2011; Westra et al., 2014). By
using groundwater as a proxy of terrestrial inputs from the watersheds of different types of lakes,
we simulated the effects of storm events and compared the sensitivity of different terrestrial
DOC sources to photodegradation. Interestingly, we found DOC from the watersheds of
oligotrophic and eutrophic lakes showed stronger changes in CDOM, compared to DOC from the
watersheds of the brown-water lakes that showed significantly larger changes in DOC quantity.
This difference may be due to the more allochthonous nature of the brown-water DOC, which is
highly photolabile, resulting in greater changes in DOC quantity due to its ability to absorb UV
radiation (Bertilsson and Tranvik, 2000). The less allochthonous and more microbially derived
DOC from the watersheds of the eutrophic and oligotrophic lakes may be less photolabile with
fewer UV-absorbing chromophores. Results of the DDA may be helpful in predicting changes in
other lakes based on their trophic status. $SUVA_{320}$ is the variable most likely to change due to
photodegradation in eutrophic and oligotrophic lakes. In contrast, DOC concentration is the
variable most likely to change in brown-water lakes due to photodegradation. Both results (DOC
and $SUVA_{320}$) highlight how lakes of varying trophic status respond to photodegradation. These
results can be used to predict how lakes not included in this study will respond to increased DOC
concentrations (i.e. browning).
Across our study lakes, changes in DIC production scaled linearly with initial
groundwater DOC concentration. Lacawac had the highest initial DOC concentration (59.4 ±
6.1) and the highest average DIC production, while Giles had the lowest initial DOC
concentration (6.0 ± 0.6) and the lowest average DIC production. This suggests that DOC





concentration plays a critical role in determining the fate of DOC (Leech et al., 2014; Lapierre et
al., 2013). Recent research has also reported that residence time controls organic carbon
decomposition across a wide range of freshwater ecosystems (Catalan et al., 2016, Evans et al.,
2017). However, extreme precipitation events may shorten the residence time of lakes,
effectively flushing out fresh DOC and preventing significant in-lake degradation from occurring
(de Wit et al., 2018). For the terrestrial DOC from the oligotrophic and eutrophic lakes, a
significant fraction was not degraded, which may mean that terrestrial inputs from these
watersheds undergoes less immediate in-lake processing and instead is exported downstream.
Our results indicate that differences in the fate and processing of DOC from the watersheds of a
range of lake types have important implications for determining which lakes may release more
$CO_2$ versus export DOC downstream (Weyhenmeyer et al., 2012; Zwart et al., 2015;
Weyhenmeyer and Conley, 2017).

Even though we observed similar responses to photodegradation in the brown-water lakes

(Fig. 1), the magnitude of the response varied and may have been related to the initial DOC
concentration. Initial concentrations (mg $L^{-1}$) of terrestrial DOC from Lacawac (59.4 ± 6.1) were
almost 3x higher than Annie (20.7 ± 0.5). Average DOC losses for both lakes due to
photodegradation were ~35%. The main difference between Lacawac and Annie was the DIC
percent change due to photodegradation (Fig. 1). Average percent increases in DIC for Lacawac
were close to 400%, whereas in Annie it was ~85%. Despite the fact that both Annie and
Lacawac are brown-water lakes, their different DIC production rates indicate that certain types of
terrestrial DOC may be more photolabile than others and capable of outgassing large amounts of
$CO_2$. The DDA analysis did also pick out the separation between Lacawac and Annie primarily
on axis 1 (DOC). The responses in Annie shared similarities with the other 3 lakes while





Lacawac only overlapped with Annie. When put in the context of the entire DOC pool for each
lake, photomineralization accounted for 2% of the carbon loss. We anticipated that terrestrial
DOC from subtropical lakes would undergo additional microbial processing due to the higher
temperatures year-round. In a comparison between boreal Swedish and tropical Brazilian lakes,
Graneli et al., (1998) also found strong similarities in changes of DOC concentrations and DIC
production between lakes from the different latitudes. A weak significant correlation between
DOC concentration and DIC production has also been observed in Amazon clear water systems
(Amado et al., 2003)

**Conclusions**

Here we showed that photodegradation can be more important than biodegradation in

processing watershed inputs of terrestrial DOC on short time scales in the surface waters of a
lake. The responses that we observed varied with lake trophic status. Quantitative changes in
DOC, DIC, and DO were strongest in the terrestrial DOC from the watersheds of the brown-
water lakes, whereas the largest changes in $SUVA_{320}$ were observed in the terrestrial DOC from
the watersheds of the eutrophic and oligotrophic lakes. Consistent with prior studies, we found
that sunlight can impact not only changes in the concentration, but also the absorbance properties
of the DOC pool. We observed a range of 1.2 to 34% of the carbon pool processed in one week.
As DOC concentrations increase in some aquatic ecosystems, the potential for increased $CO_2$
outgassing due to photo-mineralization also increases. On short time scales, sunlight had
important impacts on our study lakes. Future studies should focus on additional lakes, longer
timescales, and integrating DIC production throughout the water column.



Over the next century, DOC concentrations in northern boreal lakes are projected to
increase by 65% (Larsen et al., 2011). Thus, understanding the fate of terrestrial sourced organic
material will be essential for predicting the ecological consequences for lakes and downstream
ecosystems (Solomon et al., 2015; Williamson et al., 2015; Finstad et al., 2016). Improving
estimates of organic carbon processing in lakes will be an important component of creating more
complete carbon budgets (Hanson et al., 2004; 2014) and global estimates of $CO_2$ emissions can
be more accurately scaled to reflect the ability of lakes to act as $CO_2$ sinks or sources as
browning continues (Lapierre et al., 2013, Evans et al., 2017).

**Acknowledgements**
This project was supported by NSF grants DBI-1318747 (LBK, CEW, and others), DBI-1542085
(LBK, CEW, CMD), DEB 1754276 (CEW), the Robert Estabrook Moeller Research Fellow
Award (CMD, JAB), the Isabel and Arthur Watres Student Research Award (SM), and the staff
and resources of Archbold Biological Station. Data were provided by Archbold Biological
Station. We would like to thank Kevin Main for assistance in collecting Lake Annie water
samples and Erin Overholt for assistance in the lab and logistical support. The authors would like
to thank two anonymous reviewers for their assistance in improving the manuscript. The authors
declare no competing interests.

**Data Availability:**
Data and metadata will be made available in the Environmental Data Initiative repository. Data
archiving will be led by C. Dempsey and J. Brentrup.



**Author Contribution Statement**
CMD, JAB, and CEW designed the study with help from LBK, EEG, and HMS. CMD, JAB,
SM, and HMS collected the water samples and ran the experiments.  DPM provided the
analytical equipment for measuring DIC and DOC. CMD and JAB analyzed the data, and CMD
and MTG conducted the statistical and DDA analyses. CMD and JAB wrote the manuscript with
contributions from all of the authors.





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
