# Peer review of "Title: The relative importance of photodegradation and biodegradation of terrestrially derived"

_Biogeosciences, 2020_

## Referee Comment (RC1) · Anonymous Referee #1 · 18 Jun 2020

The manuscript presents a very interesting study aimed at discerning processes within lakes that control the mineralization of DOC, whether biodegradation or photodegradation prevails. It's an important contribution to the field to quantify the possible load of $CO_2$ emitted from lakes, in the current scenario of climate change, and studies in this respect are highly valuable. Moreover, studies addressing in-lake DOC processing as both biodegradation and photodegradation are scarce. Therefore, this manuscript is a timely contribution to this area of research. The manuscript is well written and despite being easy to follow it is not trivial, but instead of a high scientific soundness.

[Figure]

The aims are well described, the abstract is informative and well resumes the study. I have some comments that are highlighted below, and I recommend to consider, in the discussion, mechanisms of primary production in lakes and DOC lability, as this may be related to the different biodegradation response. Overall I enjoyed reading this manuscript and I would like to see it published after some revision, as suggested by the following comments.

Introduction

84: I would suggest to add a few words clarifying mineralization, degradation, respiration

95: Just a guess but in the Arctic it might be that the low temperatures slow down bacterial activity and therefore, photomineralization prevails. I think it really depends on other bio-physical factors and external climate conditions, could you provide other examples?

97: why does the source of inland water CO2 remain uncertain? What are the possible explanations besides bacterial activity and photomineralization?

104/105: I would talk a bit more on CDOM, giving some more information. As it can represent a great portion of DOC, I think it is a bit reductive to refer to it only as "absorbance characteristics".

107: do you mean the smaller MW, the less attenuation by DOC because of larger molecules? Or do you mean that light can make DOC otherwise unavailable to bacteria, available? Eg. Kieber et al., Nature 1989 (referred to marine systems, but could be probably applied to freshwater environments as well).

111: in this study, does sunlight-driven degradation accelerate subsequent bacterial activity?

115-119: this sentence is very long and hard to follow. Could you break it up into two sentences or rephrase it?

125: when you say reactive, do you mean easily broken down/degraded/uptaken by bacteria?

126: different optical properties how? Can you specify?

Methods:

183: can you specify why groundwater can be a proxy for terrestrial DOC runoff? I have seen it afterwards in the discussion but I would just add a few words here as well.

209: additionally to the 3 treatments, did you also check for combined photodegradation + biodegradation setting or would it have been too tricky to discern which process from which in that case? Maybe it could be some experiment to try in the future. What are possible limitations? You could add a few sentences in the discussion about that, if it's a feasible experiment.

239: why did you inoculate the samples with GW only for the biodegradation experiment? Would it have been interesting to see the short-term response of photodegradation too, and check which one could be faster or prevailing? Just asking for curiosity.

255: what was the temperature of the Lacawac Lake in the experimental months compared to each other lake, singularly? Was there a high difference?

277: I would explain better the meaning of Sr: how do you calculate it and that it increases with photodegradation, thus its increase is inversely related to MW. What about its relation to biodegradation instead?

299-308: it would be easier to see if presented in a suite of equations/formulas.

314: As "seasonal response" do you mean the monthly differences? June July and august seems more similar in DOC response compared to May in Lacawac lake. Could it be because of temperature? (Fig S3). It seems to me that brown water lakes with higher GW DOC background showed a more variable response. Were there any significant differences among different months?

377 and Figure 1: it seems that biodegradation mostly affected DO concentration with respect to control with no evident differences between the lakes, while it didn't change much for DOC, DIC, SUVA and Sr. Why do you think the losses of DO due to photodegradation were higher with respect to biodegradation? Again, do you think temperature may have played a role? Together with light there must have been an increase in temperature in all samples exposed to light. This could not be much evident, probably, in the biodegradation samples (slighter differences among lakes). In general T increases bacterial activity and therefore DO loss. Is it possible that lakes with higher load of particulates may have experienced higher temperatures and therefore, more changes in the DOC and DO?

Discussion

487 and 493 (DOC and DO): It seems to me that where the DOC background is lower (eutrophic and oligotrophic) the bacterial response is higher. Could the higher DOC be the result of a bacterial turnover of carbon, producing different DOC over previous substrates? If looking at Sr, Gilles and Waynewood have the lowest values, possibly indicating a production of DOM of higher MW with respect from starting compounds. Maybe the differences seem minimal but if changing the scale in figure 1e for the biodegradation part, possibly differences are more evident. Maybe you could check with S (e.g. 275-295) or CDOM absorbance to further investigate into that process, whether it could actually be a bacterial contribution. I may think that in general, where the DOC background is lower, the partially photodegraded DOC is extremely easy to be processed and taken up by bacteria, therefore it's harder to detect it. In your samples you may not see this mechanism because samples exposed to photodegradation excluded bacterial activity by setup and viceversa, but if I understood the setup right, it may be that that the photodegraded fraction in the biodegradation experiments might have quickly been assimilated by bacteria already in situ (i.e. in the lake at samples collection), remaining with a fraction to be processed and a large unprocessed fraction. Thus, complex substrates need light as a catalyst for further bacterial assimilation. And

time can also explain this process, the turnover rate for bacteria is longer than for light, depending on the complexity and reactivity of the starting material. Some more argumentation should be added in this respect (see comment below as well).

501-503: this is very interesting. Why do you think this mechanism occurs? How can it be related to primary production and the release of "fresh" and labile DOC that bacteria can assimilate more easily? I think some argumentation should be added here, and I believe it would be very useful to discuss about different carbon fractions and reactivities (labile, semi-labile, refractory)..

520: although for marine systems, I think here Kieber et al. 1989 could be added as a reference https://www.nature.com/articles/341637a0

547: it joins to my previous comment (501-503): "microbially derived" should be better explained as bacteria may channel a great portion of DOC produced by autotrophic organisms.

559: I suggest to better explain this concept: the more DOC the more DIC produced?

---

## Referee Comment (RC2) · Anonymous Referee #2 · 22 Jun 2020

Overview:

The manuscript presents data on the photo-oxidation of dissolved organic carbon (DOC) using groundwater inflow as the source waters to 4 freshwater lake ecosystems. While it is an interesting study, there are a number of issues with the experimental aspects of the work that need to be clarified at present as there is considerable variability in the DOC concentrations and the DIC yields are surprisingly low with an apparently large pool of missing carbon unaccounted for in the experimental analysis when viewed in terms of a C mass balance.

General Comments:

*Carbon mass balance and potential loss of $CO_2$ to headspace in exetainers:*

The description of the DIC analysis raises a number of questions as to how well the measurements were made in this work. Measuring changes in DIC in the presence of large concentrations of DOC has always been a challenge (Granéli *et al.*, 1998; Granéli *et al.*, 1996) but there are methods available to do this reasonably well (Porcal *et al.*, 2015). However, in the present case the method description seems to indicate that for the DIC and DO measurements there was a headspace in the Exetainer Vials used. Having a headspace when measuring dissolved gases like $O_2$ and $CO_2$ is problematic as there will be a considerable amount of gas exchange to the headspace. This is likely the reason why the DIC yield from DOC photo-oxidation are so low in this work and inconsistent with previous studies. At the very least the description has to be improved in a revised manuscript so the analytical issues arising from this type of measurement can at least be understood in a reasonable framework.

In the manuscript there is no real attempt at an overall carbon balance as there were no measurements of POC taken. This is problematic from the point of view of the overall experimental design but it is made more complicated by a misreading of the Cory et al. (2014) work by which the authors have confused $O_2$ and $CO_2$ (see below for full details) stoichiometries and this unfortunately impacts the interpretation of the results considerably – at best it is a series of typos at worst a serious misunderstanding of the earlier work and of what this paper itself was trying to achieve (i.e. why measure DIC if you can just use a relationship from elsewhere).

*Role of iron in the photochemical reduction of DOC:*

Iron and pH have been identified previously as playing an important role in the photo-oxidation of DOC in freshwaters (Gu *et al.*, 2017; Molot *et al.*, 2005), it is a pity then that there are apparently no measurements of the iron content of these waters.

Specific comments:

Line 77: An additional reference of note on the Brownification of fresh waters is the recent review by Kritzberg *et al.* (2020).

Line 179: Where does the data for the residence time of the lakes come from? It would also be useful to include the estimates of sinks/sources and lake inventory that were used in estimating the residence time. In this context it would be useful to know what the catchment sizes were and the average rainfall to each lake. This would help the reader understand more the processes impacting DOC in the lakes.

Line 199: Please indicate if the GF/F filter was pre-combusted before use to remove any DOC on the filter itself.

Line 218: Are these the usual Labco Exetainer vials? If so please provide the part number etc as these are commonly used for dissolved gas samples and so are well known to most researchers.

Line 218: So does this mean there was a 2 mL headspace in the Exetainers? This will impact the measurements of the DO and DIC considerably (Spötl, 2005; Waldron *et al.*, 2014), see also recommendations from the lab at UC Davis: https://stableisotopefacility.ucdavis.edu/dictracegassamplepreparation.html

Line 223: It would be useful to restate here that these are all groundwater samples and not water from the adjacent lake.

Line 233: This is 100 μL of groundwater to both the 35 mL Quartz tube and the 12 mL Exetainer? If this is the case how are the data then corrected for the differences in the additions between the DOC and DIC samples?

Line 233: The bacterial community in the groundwater may be significantly different from that found in the lake, most notably in the presumably the abundance of photosynthetic organisms and the response to light. Could photoinhibition of bacterial activity also have been important here?

Line 248: The samples wrapped in Al foil may have been exposed to greater temperatures during the course of the incubations due to solar heating. Some indication of the *in situ* temperatures and the solar irradiation received (e.g. the data from the radiometer) would be helpful then to gauge if this could have been an influence on the experiment.

Line 266: Please report the standards and/or Certified Reference Materials used for the DOC analysis.

Line 268: Were the DIC samples acidified through the Exetainer septum to prevent gas exchange?

Line 268: What was the volume of sulfuric acid added to each vial?

Line 271: What does well mixed mean in this case? That the headspace in the exetainers was shaken with the water layer. Where the samples acidified prior to this mixing? – see the comment also above regarding the acidification steps.

Line 298: Unfortunately this statement is incorrect as the original citation (Cory *et al.*, 2014) assumes a 0.5 mol $O_2$ consumed to 1 mol DOC oxidized for partial photooxidation, not $CO_2$ as stated in the present manuscript. The Cory et al. (2014) value is also not valid as other work has shown this value can vary depending on the river water itself (Xie *et al.*, 2004).

Line 299: What does [DIC*2] signify here? Are you suggesting that half the photo-oxidized carbon turns into some other form of carbon? As the previous sentence in the manuscript linking $CO_2$ production from DOC was erroneous, this sentence is also incorrect. It begs the question as to how the C balance is achieved if only 50% of the DOC photo-oxidized forms $CO_2$ what happens to the other 50% of the C as normally CO production is only a small pathway.

Line 301: What is this pool of carbon then, if it is converted from DOC but it is not DIC it has to then be POC by default, unless the authors are arguing for a 3rd form of dissolved carbon? See the work Porcal *et al.* (2015) for more details on the carbon balance in these types of experiments.

Line 366: Table 1 - I found some of the statistical relationships to be not credible here; given the data provided so it would be extremely useful to include more details on how the statistics were generated here and for the values to be rechecked. For example when the P/B column indicates a $p < 0.001$ values for DIC results but the data clearly overlap at the 1 or $2\sigma$ level then something is not right with regard to the p value reported: $49.1 \pm 11.4$ compared to $25.3 \pm 7.2$ and $20.4 \pm 1.9$ compared to $17.7 \pm 3.0$

Line 366: Table 1: The DIC yields from photo-oxidation are very low, for example only at Lacawac there is approximately a 30 $\mu$mol $L^{-1}$ increase in DIC for a 1500 $\mu$mol $L^{-1}$ decrease in DOC so either there is very large POC production or there is something very wrong with the DIC or DOC values. Given the issues noted above for the DIC measurements it is most likely those values that are questionable.

Line 490: The red precipitate is likely iron oxide but was this found in the controls as well?

Line 491: If there is a precipitate then the iron is not being released 'back into the water', it is now precipitating out into the solid phase as the complexing agents responsible for solubilizing it have been destroyed.

Line 495: Sloppy feeding is a term normally applied to zooplankton grazing on bacteria and not bacteria themselves. Why mention this in the context of $O_2$? As producing DOC does nothing to the $O_2$ content necessarily – the $O_2$ is used up in respiration.

References:

Cory, R.M., Ward, C.P., Crump, B.C., Kling, G.W., 2014. Sunlight controls water column processing of carbon in arctic fresh waters. Science 345 (6199), 925-928.

Granéli, W., Lindell, M., de Faria, B.M., de Assis Esteves, F., 1998. Photoproduction of dissolved inorganic carbon in temperate and tropical lakes – dependence on wavelength band and dissolved organic carbon concentration. Biogeochemistry 43 (2), 175-195.

Granéli, W., Lindell, M., Tranvik, L., 1996. Photo-oxidative production of dissolved inorganic carbon in lakes of different humic content. Limnology And Oceanography 41 (4), 698-706.

Gu, Y., Lensu, A., Perämäki, S., Ojala, A., Vähätalo, A.V., 2017. Iron and pH Regulating the Photochemical Mineralization of Dissolved Organic Carbon. ACS Omega 2 (5), 1905-1914.

Kritzberg, E.S., Hasselquist, E.M., Škerlep, M., Löfgren, S., Olsson, O., Stadmark, J., Valinia, S., Hansson, L.-A., Laudon, H., 2020. Browning of freshwaters: Consequences to ecosystem services, underlying drivers, and potential mitigation measures. AMBIO 49 (2), 375-390.

Molot, L.A., Hudson, J.J., Dillon, P.J., Miller, S.A., 2005. Effect of pH on photo-oxidation of dissolved organic carbon by hydroxyl radicals in a coloured, softwater stream. Aquatic Sciences 67 (2), 189-195.

Porcal, P., Dillon, P.J., Molot, L.A., 2015. Temperature Dependence of Photodegradation of Dissolved Organic Matter to Dissolved Inorganic Carbon and Particulate Organic Carbon. PLoS ONE 10 (6), e0128884.

Spötl, C., 2005. A robust and fast method of sampling and analysis of δ13C of dissolved inorganic carbon in ground waters. Isotopes in Environmental and Health Studies 41 (3), 217-221.

Waldron, S., Marian Scott, E., Vihermaa, L.E., Newton, J., 2014. Quantifying precision and accuracy of measurements of dissolved inorganic carbon stable isotopic composition using continuous-flow isotope-ratio mass spectrometry. Rapid Communications in Mass Spectrometry 28 (10), 1117-1126.

Xie, H., Zafiriou, O.C., Cai, W.-J., Zepp, R.G., Wang, Y., 2004. Photooxidation and Its Effects on the Carboxyl Content of Dissolved Organic Matter in Two Coastal Rivers in the Southeastern United States. Environmental Science & Technology 38 (15), 4113-4119.

---

## Author Comment (AC1) · 2 Jul 2020

Thank you for the comments and suggestions. We will work to incorporate them in a revised manuscript. Below I have replied to specific questions/statements using line numbers from the original manuscript.

Line 84: This can be added

Line 95: To our knowledge there are only a handful of paired bio and photodegradation studies. It is certainly possible that colder temperatures slow the rate of biodeogradation in the Arctic. The length of photodegradation experiments are important. Sunlight acts much more rapidly than bacteria.

Line 97: This is related to the uncertainty in organic carbon inputs, processing, and burial within aquatic ecosystems on regional or global scales. Butman et al, 2016 provides an explanation of some of these reasons.

Line 104/105: We can change our reference to optical/absorbance characteristics to CDOM.

Line 107: No. We were trying to point out that sunlight effects DOC in other ways (i.e. other than mineralization).

Line 125: Yes. We could have phrased that to be clearer.

Line 126: In the papers we cited, the optical properties were different between forested and disturbed study sites.

Line 183: Yes. That can be added

Line 209: We did not include it in this manuscript but we did run a combined photodegradation and biodegradation treatment. The response in those treatment was similar to the photodegradation only treatments, but greater (i.e. more DOC lost or more DIC produced). This would certainly be something to explore further, but we did not test microbial abundance at the beginning and of those treatments. It is possible that UV exposure lysed cells, which were more susceptible to photodegradation.

Line 239: For the biodegradation experiments, we wanted to provide a fresh source of bacteria. In hindsight, we should have used lake water, but we only had groundwater shipped from Annie. For consistency, we opted to use groundwater for all experiments. We did not test the short term response to DOC here, but since 2016 we have run additional experiments since 2016 to test out shorter time frames (i.e. 48 hours). Neat suggestion.

Line 255: The average temperature (measured at 0.1m) for each 7 day experiment in Lacawac was as follows: May- 15.1, June 20.1, July- 25.4, August 26.2. All are degrees Celsius.

Line 277: This can be added

Line 299-308: Good suggestion. This can be incorporated.

Line 314: Yes. Monthly differences is the same as seasonal response. It could be lake temperature, but it may have more to do with how long it took to filter the Lacawac groundwater. Annie water was shipped to us each month and there is not a strong seasonal response in that lake (SI Fig 3). For Lacawac, the groundwater was collected in early May, but we were unable to filter the remaining water until ∼10 days later. There was a large amount of particulates in the samples which may have continued to degrade. It is not reported in the manuscript but for Lacawac starting concentration of DOC in May, June, July, and August was ∼48, ∼63, ∼63, and ∼61 mg/L. While we did not assess the role of temperature, the starting concentration of DOC is important. Temperature does influence the photodegradation process.

Line 377: It is possible temperature played a role. One the timescales in which we conducted the experiments, photodegradation acts much more rapidly on the DOC pool when compared to bacteria.

Line 487-493: We kept the treatments separate in this manuscript and did not do a combined photo and biodegradation treatment. We also don't know if a combined treatment would kill or lyse bacteria due to UV exposure. Samples were kept at the surface of the lake for 7 days. It would be interesting to test this to determine whether bacteria can survive for 7 days. We do think that the length of time experiments are run is important.

Line 501-503: We think this occurs because the properties of the watershed influence the lake. Lacawac and Annie have bog areas that contribute to their brown water

description. There is likely a fraction of the DOC pool in each lake that is capable of being biodegraded, a fraction that is capable of being photodegraded, and a fraction that not able to be processed. More of this can be added to the discussion.

References: Butman, D., Stackpoole, S., Stets, E., McDonal, C.P., Clow, D.W., Striegl, R.G. 2016. Aquatic carbon cycling in the conterminous United State and implications for terrestrial carbon accounting. PNAS: 113(1), 58-63.

---

## Author Comment (AC2) · 7 Jul 2020

Thank you for the suggestions and feedback. Our future work with DIC will be sure to acidify immediately and remove the headspace issue. We still think the DIC results are relevant as samples were mixed prior to analysis. It does seem likely that some of the $CO_2$ partitioned into the headspace. The results we present here represent minima for DIC production due to photodegradation. I re-ran the calculations (assuming 1:1 molar ratio) using the oxygen data. Those results are presented in the attached figure. While there was 2 mL of headspace in the oxygen exetainer vials, the Winkler

method was carried out inside those vials and they were mixed multiple times throughout the process. Using the oxygen data, we are still likely underestimating the carbon pool processed by sunlight. Allesson et al (2016) suggest an average RQ value for photodegration of 3.5.

We did not attempt to develop an overall carbon balance as we did not measure POC. I did make a mistake in regards to calculating the amount of carbon photomineralized. Thank you for finding my error.

We did not collect iron data as a part of this experiment. Other researchers at the lake collected samples from the three Pocono lakes and from wetlands surrounding each lake in 2018 and 2019. The concentrations they reported were low (unpublished data). Giles Surface water: 0.025 mg/L, Lacawac Surface water: 0.122 mg/L, and Waynewood Surface water: 0.13 mg/L. Giles wetland: 0.22 mg/L, Lacawac wetland: 4.7 mg/L, and Waynewood wetland: 0 mg/L.

Below we have responded to specific comments not addressed above:

Line 179: This information is in the Moeller et al 1995 paper we cited. We could pull more of that information into the manuscript.

Line 199: All filters were pre-combusted.

Line 218: Yes. Labco Exetainer vials 138W.

Line 218: Yes. 2mL of headspace. The samples (i.e water and headspace) in each vial were mixed for 30 seconds. A syringe was used to carefully extract 5 mL of sample. Samples were then acidified using 200 uL of 0.1N H2SO4. 5 mL of nitrogen gas was then added to the syringe and the syringe was then mixed prior to affixing to the GC.

Line 233: The data are not corrected for difference in volumes.

Line 248: We have in situ lake temperature for Lacawac, but we did not install temperature sensors inside of aluminum foil wrapped tubes. We do have radiometer data

available that could be added.

Line 266: We use a 50ppm TOC standard from Aqua Solutions and create dilutions to calibrate the TOC analyzer.

Line 366: These will be double checked.

Line 490: No. The red precipitate only occurs in the samples exposed to sunlight.

Line 491: The sentence will be re-worded to be more clear. We were suggesting a potential reason as to why DOC concentrations increased in Giles and Waynewood. The intent was to refer to the DOC being released back into the water as the iron precipitated out.

References: Allesson, L., L. Ström, and M. Berggren(2016), Impact of photochemical processing of DOC on the bacterioplankton respiratory quotient in aquatic ecosystems, Geophys. Res. Lett., 43, 7538–7545, doi:10.1002/2016GL069621.

[Figure]

[Figure]

**Fig. 1.** Figure 2 based off of oxygen data

---

## Author Response (AR1)

[revised manuscript text omitted]

of Lake Lacawac during each 7 day experiment

| Month | Dates | Total Light (J km$^{-2}$ nm$^{-1}$) | Mean Surface Temp (° C) ± SD |
|-------|-------|------|------|
| May | May 9-16 | 299.9 | 15.1 ± **1.5** |
| June | June 7-13 | 365.4 | 20.1 ± **1.4** |
| July | July 7-13 | 320.9 | 25.4 ± **0.8** |
| August | August 8-14 | 365.4 | 26.2 ± **1.1** |

[Figure]

**SFigure 1.** Percent transmittance of the quartz and borosilicate glass that was used for the
experiments.
**Supplemental Methods**

The photodegradation treatments described above (DOC, DIC, DO, $S_r$, and SUVA$_{320}$) were normalized to the total amount of light received by the samples for each month. This allowed us to determine the impact of seasonality. To calculate the total amount of light (J m$^{-2}$ nm$^{-1}$), a modeled solar spectrum (280-700nm) was created. The base spectrum was generated with the

Quick TUV Calculator (version 5.2; http://cprm.acom.ucar.edu/Models/TUV/Interactive_TUV/)

for June 21 through June 27, 2016 (Madronich 1993). The latitude and longitude of Lake

Lacawac (Table 1) was provided and the ozone concentration from the Total Ozone Mapping

Spectrometer (TOMS; https://ozoneaq.gsfc.nasa.gov/tools/ozonemap/) for each day was entered.

We then fit this modeled solar spectrum to our GUV data for each experimental timeframe using

Solver in Microsoft Excel (version 2013). A best fit was determined by calculating the square of the difference between the measured GUV data and the values estimated by the model for the

305 and 340nm wavelengths. In the resulting modeled solar spectra (SI Fig. 2), the total amount of light (280-700nm) was summed for each month of the experiment and was used to standardize the concentration and optical data described above.

[Figure]

**SFigure 2.** Modeled solar spectra for each month plotted against wavelength (nm)

[Figure]

**SFigure 3**. Normalized photodegradation data (described in SI Fig 2) for each variable, lake, and
month. Photodegradation samples were compared to controls (0 value on each panel). The panels
are arranged as follows: A) DOC, B) DIC, C) DO, D) SUVA$_{320}$, and E) S$_r$. Statistical
significance is indicated by the letter(s) above each bar. Months were compared using an
ANOVA with a Tukey post-hoc test (CI = 95%). n = 3 for each bar.

 **Response to reviewer comments**

The authors would like to thank the two anonymous reviewers for their thoughtful comments.
We have incorporated many of the suggestions into the revised draft of the manuscript. Below
we have provided specific comments in "red" too each line item.

**Anonymous Referee #1**

The manuscript presents a very interesting study aimed at discerning processes within lakes that
control the mineralization of DOC, whether biodegradation or photodegradation prevails. It's an
important contribution to the field to quantify the possible load of CO2 emitted from lakes, in the
current scenario of climate change, and studies in this respect are highly valuable. Moreover,
studies addressing in-lake DOC processing as both biodegradation and photodegradation are
scarce. Therefore, this manuscript is a timely contribution to this area of research. The
manuscript is well written and despite being easy to follow it is not trivial, but instead of a high
scientific soundness.

The aims are well described, the abstract is informative and well resumes the study.
I have some comments that are highlighted below, and I recommend to consider, in the
discussion, mechanisms of primary production in lakes and DOC lability, as this may be related
to the different biodegradation response. Overall I enjoyed reading this manuscript and I would
like to see it published after some revision, as suggested by the following comments.

Introduction

84: I would suggest to add a few words clarifying mineralization, degradation, respiration
These were added into the introduction.

95: Just a guess but in the Arctic it might be that the low temperatures slow down
bacterial activity and therefore, photomineralization prevails. I think it really depends
on other bio-physical factors and external climate conditions, could you provide other
examples?
There are so few studies that directly compare photo and biodegradation. The amount of time
experiments are run appears to be important. Sunlight degrades material fairly quickly, whereas
microbes need more time.

97: why does the source of inland water CO2 remain uncertain? What are the possible
explanations besides bacterial activity and photomineralization?
Some of this is land use, but most of the uncertainty is due to the lack of measurements in
different aquatic ecosystems.

104/105: I would talk a bit more on CDOM, giving some more information. As it can
represent a great portion of DOC, I think it is a bit reductive to refer to it only as "absorbance
characteristics".
We went back and forth on this in earlier drafts, but ended up using the term CDOM throughout
the revised draft. We added some description into the manuscript.

107: do you mean the smaller MW, the less attenuation by DOC because of larger
molecules? Or do you mean that light can make DOC otherwise unavailable to bacteria,
available? Eg. Kieber et al., Nature 1989 (referred to marine systems, but could be
probably applied to freshwater environments as well).
No. We are just trying to point out that sunlight effects DOC in other ways (i.e. not just
mineralization)

111: in this study, does sunlight-driven degradation accelerate subsequent bacterial
activity?
We unfortunately did not test that in this study.

115-119: this sentence is very long and hard to follow. Could you break it up into two
sentences or rephrase it?
Yes.
125: when you say reactive, do you mean easily broken down/degraded/uptaken by
bacteria?
Yes. More easily degraded or broken down. We could have phrased that to be clearer.

126: different optical properties how? Can you specify?
In the papers that were cited, the optical properties (i.e. $SUVA_{254}$, $S_r$, etc...) were different
between forested and disturbed study sites.

Methods:

183: can you specify why groundwater can be a proxy for terrestrial DOC runoff? I
have seen it afterwards in the discussion but I would just add a few words here as well.
Yes.

209: additionally to the 3 treatments, did you also check for combined photodegradation
+ biodegradation setting or would it have been too tricky to discern which process
from which in that case? Maybe it could be some experiment to try in the future. What
are possible limitations? You could add a few sentences in the discussion about that, if
it's a feasible experiment.
We did not include it in this manuscript, but we did run a combined photodegradation and
biodegradation treatment. The response in those treatments was similar to the photodegradation
response, but greater. Those treatments did not fit well with the points we wanted to make here.

239: why did you inoculate the samples with GW only for the biodegradation experiment?
Would it have been interesting to see the short-term response of photodegradation
too, and check which one could be faster or prevailing? Just asking for curiosity.
For the biodegradation experiments, we wanted to provide a fresh source of bacteria. In
hindsight, we should have used bacteria from the surface each lake, but we only had groundwater
shipped from Lake Annie. For consistency, we opted to use groundwater for all biodegradation
treatments.

We did not test the short-term response in this set of experiments, but we have run additional
experiments since 2016 to test DOC photodegradation over 48 hours and over 90 days.
255: what was the temperature of the Lacawac Lake in the experimental months compared
to each other lake, singularly? Was there a high difference?
The experiments were run only in Lacawac.  Average lake temperature for each 7 day
experiment has been added to SI Table 1.
277: I would explain better the meaning of Sr: how do you calculate it and that it
increases with photodegradation, thus its increase is inversely related to MW. What
about its relation to biodegradation instead?
The wavelengths used for calculated the ratio have been added.  The full description is in the
Helms 2008 paper.
299-308: it would be easier to see if presented in a suite of equations/formulas.
Good suggestion. These were added.
314: As "seasonal response" do you mean the monthly differences? June July and august
seems more similar in DOC response compared to May in Lacawac lake. Could
it be because of temperature? (Fig S3). It seems to me that brown water lakes with
higher GW DOC background showed a more variable response. Were there any significant
differences among different months?
Yes.  Monthly differences is the same as seasonal response.  It could be lake temperature, but it
may have more to do with how long it took to filter the Lacawac groundwater.  Annie water was
shipped to us each month and there is not a strong seasonal response in that lake (SI Fig 3).  For
Lacawac, the groundwater was collected in early May, but we were unable to filter the remaining
water until ~10 days later.  There was a large amount of particulates in the samples which may
have continued to degrade.  It is not reported in the manuscript but for Lacawac the starting
concentration of DOC in May, June, July, and August was ~48, ~63, ~63, and ~61 mg/L.  While
we did not assess the role of temperature, the starting concentration of DOC is important.
377 and Figure 1: it seems that biodegradation mostly affected DO concentration with
respect to control with no evident differences between the lakes, while it didn't change
much for DOC, DIC, SUVA and Sr. Why do you think the losses of DO due to photodegradation
were higher with respect to biodegradation? Again, do you think temperature
may have played a role? Together with light there must have been an increase
in temperature in all samples exposed to light. This could not be much evident, probably,
in the biodegradation samples (slighter differences among lakes). In general T
increases bacterial activity and therefore DO loss. Is it possible that lakes with higher
load of particulates may have experienced higher temperatures and therefore, more
changes in the DOC and DO?
It is possible that temperature played a role in the seasonal differences.  Since all experiments
were conducted in Lake Lacawac, all treatments were at the same temperature each month.  On
the timescales in which we conducted the experiments, photodegradation acts on the DOC much
more rapidly than biodegradation.  Given longer time periods, bacteria likely become important
in the degradation of the DOC.  Porcal et al (2015) showed that temperature is important to photodegradation. Higher temperatures caused larger losses of DOC compared to colder
temperatures.
Discussion
487 and 493 (DOC and DO): It seems to me that where the DOC background is lower
(eutrophic and oligotrophic) the bacterial response is higher. Could the higher DOC
be the result of a bacterial turnover of carbon, producing different DOC over previous
substrates? If looking at Sr, Gilles and Waynewood have the lowest values, possibly
indicating a production of DOM of higher MW with respect from starting compounds.
Maybe the differences seem minimal but if changing the scale in figure 1e for the
biodegradation part, possibly differences are more evident. Maybe you could check
with S (e.g. 275-295) or CDOM absorbance to further investigate into that process,
whether it could actually be a bacterial contribution. I may think that in general, where
the DOC background is lower, the partially photodegraded DOC is extremely easy to
be processed and taken up by bacteria, therefore it's harder to detect it. In your samples
you may not see this mechanism because samples exposed to photodegradation
excluded bacterial activity by setup and viceversa, but if I understood the setup right, it
may be that that the photodegraded fraction in the biodegradation experiments might
have quickly been assimilated by bacteria already in situ (i.e. in the lake at samples
collection), remaining with a fraction to be processed and a large unprocessed fraction.
Thus, complex substrates need light as a catalyst for further bacterial assimilation. And time can
also explain this process, the turnover rate for bacteria is longer than for light,
depending on the complexity and reactivity of the starting material. Some more argumentation
should be added in this respect (see comment below as well).
We kept the treatments separate in this manuscript and did not do a combined photo and
biodegradation treatment. We also don't know if a combined treatment would kill or lyse
bacteria due to UV exposure. Samples were kept at the surface of the lake for 7 days. It would
be interesting to test this to determine whether bacteria can survive for 7 days. We do think that
the length of time experiments are run is important.
501-503: this is very interesting. Why do you think this mechanism occurs? How
can it be related to primary production and the release of "fresh" and labile DOC that
bacteria can assimilate more easily? I think some argumentation should be added
here, and I believe it would be very useful to discuss about different carbon fractions
and reactivities (labile, semi-labile, refractory)..
We think this occurs because the properties of the watershed control the lake. Lacawac and
Annie have bog/wetland areas that contribute to their brown water description. There is likely a
fraction of the DOC pool in each lake that is capable of being biodegraded, a fraction that is
capable of being photodegraded, and a fraction that not able to be processed. Photodegradation
and biodegradation are both important in aquatic ecosystems and the combination of both
appears to more than each individually.
520: although for marine systems, I think here Kieber et al. 1989 could be added as a
reference https://www.nature.com/articles/341637a0
Thank you.

547: it joins to my previous comment (501-503): "microbially derived" should be better
explained as bacteria may channel a great portion of DOC produced by autotrophic
organisms.
559: I suggest to better explain this concept: the more DOC the more DIC produced?
We restated this in the revised draft. In our data, the lakes with higher concentrations of initial
DOC correspondingly saw the largest changes in DIC (i.e. Lacawac and Annie). The lakes with
the lowest initial DOC concentration saw the least amount of DIC produced during
photodegradation.
**Reviewer 2 comments**
Overview:
The manuscript presents data on the photo-oxidation of dissolved organic carbon (DOC) using
groundwater inflow as the source waters to 4 freshwater lake ecosystems. While it is an
interesting study, there are a number of issues with the experimental aspects of the work that
need to be clarified at present as there is considerable variability in the DOC concentrations and
the DIC yields are surprisingly low with an apparently large pool of missing carbon unaccounted
for in the experimental analysis when viewed in terms of a C mass balance.
General Comments:
*Carbon mass balance and potential loss of CO2 to headspace in exetainers:*
The description of the DIC analysis raises a number of questions as to how well the
measurements were made in this work. Measuring changes in DIC in the presence of large
concentrations of DOC has always been a challenge (Granéli *et al.*, 1998; Granéli *et al.*, 1996)
but there are methods available to do this reasonably well (Porcal *et al.*, 2015). However, in the
present case the method description seems to indicate that for the DIC and DO measurements
there was a headspace in the Exetainer Vials used. Having a headspace when measuring
dissolved gases like O2 and CO2 is problematic as there will be a considerable amount of gas
exchange to the headspace. This is likely the reason why the DIC yield from DOC photo-
oxidation are so low in this work and inconsistent with previous studies. At the very least the
description has to be improved in a revised manuscript so the analytical issues arising from this
type of measurement can at least be understood in a reasonable framework.
Thank you for the suggestions. Our future work with DIC will be sure to acidify immediately
and remove the headspace issue. We still think the DIC results are relevant as samples were
mixed prior to analysis. It does seem likely that some of the CO₂ partitioned into the headspace.
The results we present here represent minima for DIC production due to photodegradation.
In the manuscript there is no real attempt at an overall carbon balance as there were no
measurements of POC taken. This is problematic from the point of view of the overall
experimental design but it is made more complicated by a misreading of the Cory et al. (2014)
work by which the authors have confused O2 and CO2 (see below for full details)
stoichiometries and this unfortunately impacts the interpretation of the results considerably – at
best it is a series of typos at worst a serious misunderstanding of the earlier work and of what this paper itself was trying to achieve (i.e. why measure DIC if you can just use a relationship from
elsewhere).
Correct. Our attempt was not develop an overall carbon balance. We did not measure POC. I
did make a mistake in the calculations in regards to calculating the amount of carbon
photomineralized. Thank you for finding my error.
*Role of iron in the photochemical reduction of DOC:*
Iron and pH have been identified previously as playing an important role in the photo-oxidation
of DOC in freshwaters (Gu *et al.*, 2017; Molot *et al.*, 2005), it is a pity then that there are
apparently no measurements of the iron content of these waters.
We did not collect iron data as a part of this experiment. Other researchers at the lake collected
samples from the three Pocono lakes and from wetlands surrounding each lake in 2018. The
concentrations they reported were low (unpublished data).
Specific comments:
Line 77: An additional reference of note on the Brownification of fresh waters is the recent
review by Kritzberg *et al.* (2020).
Thank you.
Line 179: Where does the data for the residence time of the lakes come from? It would also be
useful to include the estimates of sinks/sources and lake inventory that were used in estimating
the residence time. In this context it would be useful to know what the catchment sizes were and
the average rainfall to each lake. This would help the reader understand more the processes
impacting DOC in the lakes.
The citations for the residence time have been added and details about the estimates can be found
there. Precipitation data references have also been included.
Line 199: Please indicate if the GF/F filter was pre-combusted before use to remove any DOC on
the filter itself.
All filters were pre-combusted at 450 ºC prior to filtering.
Line 218: Are these the usual Labco Exetainer vials? If so please provide the part number etc as
these are commonly used for dissolved gas samples and so are well known to most researchers.
They are the Labco Exetainer 138W vials.
Line 218: So does this mean there was a 2 mL headspace in the Exetainers? This will impact the
measurements of the DO and DIC considerably (Spötl, 2005; Waldron *et al.*, 2014), see also
recommendations from the lab at UC Davis:
https://stableisotopefacility.ucdavis.edu/dictracegassamplepreparation.html
Yes. There was 2 mL of headspace. We aired on the side of safety with not overflowing the
vials with mercury chloride. The samples water and headspace in each vial were mixed for 30
seconds. A syringe was used to carefully extract 5 mL of sample. Samples were acidified using
200 uL of 0.1N H2SO4. 5 mL of nitrogen gas was then added to the syringe and the syringe was
then mixed prior to affixing to the GC.

Line 223: It would be useful to restate here that these are all groundwater samples and not water
from the adjacent lake.
Agreed.
Line 233: This is 100 μL of groundwater to both the 35 mL Quartz tube and the 12 mL
Exetainer? If this is the case how are the data then corrected for the differences in the additions
between the DOC and DIC samples?
The data are not corrected for differences in the volumes.
Line 233: The bacterial community in the groundwater may be significantly different from that
found in the lake, most notably in the presumably the abundance of photosynthetic organisms
and the response to light. Could photoinhibition of bacterial activity also have been important
here?
Yes.  This is possible and can be added here.
Line 248: The samples wrapped in Al foil may have been exposed to greater temperatures during
the course of the incubations due to solar heating. Some indication of the *in situ* temperatures
and the solar irradiation received (e.g. the data from the radiometer) would be helpful then to
gauge if this could have been an influence on the experiment.
We have *in situ* lake temperature for Lacawac, but we did not install temperature sensors inside
of aluminum foil wrapped tubes.  We also have radiometer data that is publically available.
Line 266: Please report the standards and/or Certified Reference Materials used for the DOC
analysis.
We use a 50ppm TOC standard from Aqua Solutions and create dilutions to calibrate the TOC
analyzer.
Line 268: Were the DIC samples acidified through the Exetainer septum to prevent gas
exchange?
No.  DIC samples were acidified in a syringe after extraction from the Exetainer (see above
description)
Line 268: What was the volume of sulfuric acid added to each vial?
After removal from the exetainer, 200 μL of sulfuric acid was added to the syringe (see above
description).
Line 271: What does well mixed mean in this case? That the headspace in the exetainers was
shaken with the water layer. Where the samples acidified prior to this mixing? – see the
comment also above regarding the acidification steps.
Yes.  The headspace was mixed with the water layer prior to being acidified.
Line 298: Unfortunately this statement is incorrect as the original citation (Cory *et al.*, 2014)
assumes a 0.5 mol O2 consumed to 1 mol DOC oxidized for partial photooxidation, not CO2 as
stated in the present manuscript. The Cory et al. (2014) value is also not valid as other work has
shown this value can vary depending on the river water itself (Xie *et al.*, 2004).
We fixed this issue in the revised manuscript.

Line 299: What does [DIC*2] signify here? Are you suggesting that half the photo-oxidized
carbon turns into some other form of carbon? As the previous sentence in the manuscript linking
CO2 production from DOC was erroneous, this sentence is also incorrect. It begs the question as
to how the C balance is achieved if only 50% of the DOC photo-oxidized forms CO2 what
happens to the other 50% of the C as normally CO production is only a small pathway.
The calculations have been corrected. A revised Figure 2 has been added to the manuscript.
Line 301: What is this pool of carbon then, if it is converted from DOC but it is not DIC it has to
then be POC by default, unless the authors are arguing for a 3rd form of dissolved carbon? See
the work Porcal *et al.* (2015) for more details on the carbon balance in these types of
experiments.
We did not plan on putting together a carbon budget. The intent of Figure 2 was to show what
happens to the DOC in each lake.
Line 366: Table 1 - I found some of the statistical relationships to be not credible here; given the
data provided so it would be extremely useful to include more details on how the statistics were
generated here and for the values to be rechecked. For example when the P/B column indicates a
$p < 0.001$ values for DIC results but the data clearly overlap at the 1 or 2σ level then something
is not right with regard to the p value reported: $49.1 \pm 11.4$ compared to $25.3 \pm 7.2$ and $20.4 \pm 1.9$
compared to $17.7 \pm 3.0$
There is some variability in the data since we opted to combine all months together. We re-ran
the stats analysis and discovered some minor errors. Those have been corrected in the revised
manuscript. The errors do not change the interpretation of the data.
Line 366: Table 1: The DIC yields from photo-oxidation are very low, for example only at
Lacawac there is approximately a 30 μmol L-1 increase in DIC for a 1500 μmol L-1 decrease in
DOC so either there is very large POC production or there is something very wrong with the DIC
or DOC values. Given the issues noted above for the DIC measurements it is most likely those
values that are questionable.
Agreed. The headspace is likely the issue.
Line 490: The red precipitate is likely iron oxide but was this found in the controls as well?
No. The red precipitate only occurs in the samples exposed to sunlight.
Line 491: If there is a precipitate then the iron is not being released 'back into the water', it is
now precipitating out into the solid phase as the complexing agents responsible for solubilizing it
have been destroyed.
The sentence will be re-worded to be more clear. We were suggesting a potential reason as to
why DOC concentrations increased in Giles and Waynewood. The intent was to refer to the
DOC being released back into the water as the iron precipitated out.
Line 495: Sloppy feeding is a term normally applied to zooplankton grazing on bacteria and not
bacteria themselves. Why mention this in the context of O2? As producing DOC does nothing to
the O2 content necessarily – the O2 is used up in respiration.

This sentence can be reworded to be clearer. We were proposing respiration as mechanism for
the loss of DO in the biodegradation experiments.

---

## Referee Report (RR1)

Comments on bg-2020-manuscript-version4

The authors have made a good effort in revising their manuscript and it is significantly improved. The issue of the headspace in the samples taken for DIC and $O_2$ does limit the impact of the overall work but I am loath to reject the manuscript on this basis for 2 reasons; (i) the same procedure was applied throughout the experimental design thus the data is still comparable (ii) it provides an opportunity to highlight the importance of the ramifications of including the headspace in future research of this type as it other works have not been clear about this issue in the past and it may be a wider practice in this research community to leave a headspace in the samples without being aware of the implications of this. It is noted that for BGD as the reviews are online then the community would gradually become aware of this issue even if the paper is not ultimately deemed publishable.

Line 226: What exactly were the safety concerns about using mercury chloride that gave rise to the 2 ml headspace? As it isn't explained below despite the statement on this line. It should be made clearer the dangers of using mercury and exactly how the sampling design came about. Was it the risk from overflowing the exetainer? It should also be reported how this procedure was done in the field or lab, as it should be all performed within a tray so spills are contained. Was this procedure carried out with a pipette or a syringe?

Line 250: Is there a possibility that the presence of such a high concentration of $Hg^{2+}$ could have catalysed photoreduction of the CDOM (Luo et al., 2020)? This might lead to greater CDOM losses, it does not seem to have been considered in other works so it is hard to judge and the role or iron CDOM complexes is likely more important here but it is worth considering. It would help here to also explain in more detail how the mercury chloride impacted the CDOM measurements. Mercury chloride has been shown previously to have an absorption maximum around 305 nm (as seen for example in (Dash and Das, 2016)). This obviously would impact any optical measurements and in this case would also increase the amount of photons absorbed in the samples amended with Hg compared to those without and this facet of the work has not been commented on before it seems. It would be good to discuss then a possible alternative to mercury chloride that was optically clear.

Line 250: Information on the headspace in the bottles should be included here.

Line 254: It is not clear from the text if there is a headspace in the quartz tubes as well as for the exetainers (borosilicate vials). This information could be added here so that it is immediately clear how the experimental treatments differed.

Line 259: as for the previous comment.

Line 399: The inclusion of a headspace in the exetainers will also have reduced the apparent oxygen consumption in the samples by bringing introducing oxygen from the air into the solution contained in the exetainer. Thus, likely some of these samples may have been significantly lower in dissolved oxygen at the time of sampling. Handling of the samples (mixing) etc would also have been critical.

Line 532: As noted above the inclusion of a headspace in the exetainers will also have reduced the apparent oxygen consumption in the samples and then coupled with this approach will lead to a further underestimation of the DOC photo remineralization.

References cited:

Dash, H.R., Das, S., 2016. Interaction between mercuric chloride and extracellular polymers of biofilm-forming mercury resistant marine bacterium Bacillus thuringiensis PW-05. RSC Advances 6, 109793-109802.

Luo, H., Cheng, Q., Pan, X., 2020. Photochemical behaviors of mercury (Hg) species in aquatic systems: A systematic review on reaction process, mechanism, and influencing factor. Science of The Total Environment 720, 137540.

---

## Author Response (AR2)

[revised manuscript text omitted]

**Response to Reviewer Comments**

The authors would like to thank the reviewer for taking the time to provide comments and improve the quality of the manuscript. We recognize the headspace issue is less than ideal, but still think the main points are relevant and the data should be published. Answers to each line items are detailed below in red. The reviewer comments were appreciated.

Comments on bg-2020-manuscript-version4

Line 226: What exactly were the safety concerns about using mercury chloride that gave rise to the 2 ml headspace? As it isn't explained below despite the statement on this line. It should be made clearer the dangers of using mercury and exactly how the sampling design came about. Was it the risk from overflowing the exetainer? It should also be reported how this procedure was done in the field or lab, as it should be all performed within a tray so spills are contained. Was this procedure carried out with a pipette or a syringe?

The main concerns were the toxicity and corrosiveness of handling $HgCl_2$. I (CMD) wanted to avoid any chance of spilling in the lab as it is a mixed use lab that supports undergraduate students, graduate students, and other faculty researchers. In addition, part of the field station is open to the public. Safety was my key concern when I made the decision. We were generally following the methods laid out in Cory et al (2014). Mercury chloride was added with a pipette.

Line 250: Is there a possibility that the presence of such a high concentration of Hg2+ could have catalysed photoreduction of the CDOM (Luo et al., 2020)? This might lead to greater CDOM losses, it does not seem to have been considered in other works so it is hard to judge and the role or iron CDOM complexes is likely more important here but it is worth considering. It would help here to also explain in more detail how the mercury chloride impacted the CDOM measurements. Mercury chloride has been shown previously to have an absorption maximum around 305 nm (as seen for example in (Dash and Das, 2016)). This obviously would impact any optical measurements and in this case would also increase the amount of photons absorbed in the samples amended with Hg compared to those without and this facet of the work has not been commented on before it seems. It would be good to discuss then a possible alternative to mercury chloride that was optically clear.

This may be possible. We did not use the 1% $HgCl_2$ in our CDOM samples. CDOM scans were generated on water that was sterile filtered. We added text to clarify our own observations from pre-experiments as to how $HgCl_2$ impacted the absorbance scans in a subset of samples.

You do bring up a really good point though. Since the absorbance is higher with $HgCl_2$, more photons are being absorbed by the dissolved organic carbon. It does bring up the possibility that when $HgCl_2$ is used in these types of experiments that more DIC is produced (or DO consumed).

The only other microbial inhibitor we explored was sodium azide (Osburn et al, 2001). It is also toxic.

Line 250: Information on the headspace in the bottles should be included here.
This was added.

Line 254: It is not clear from the text if there is a headspace in the quartz tubes as well as for the exetainers (borosilicate vials). This information could be added here so that it is immediately clear how the experimental treatments differed.

There was headspace in the quartz tubes (5 mL)

Line 259: as for the previous comment.

Line 399: The inclusion of a headspace in the exetainers will also have reduced the apparent oxygen consumption in the samples by bringing introducing oxygen from the air into the solution contained in the exetainer. Thus, likely some of these samples may have been significantly lower in dissolved oxygen at the time of sampling. Handling of the samples (mixing) etc would also have been critical.

This is possible. All treatments were prepared in the same manner and then analyzed together. We compared our treatments to the control samples for each lake, which also had 2 mL of headspace. The amount of oxygen in the control samples was subtracted from the treatments so that we could record the amount consumed.

Line 532: As noted above the inclusion of a headspace in the exetainers will also have reduced the apparent oxygen consumption in the samples and then coupled with this approach will lead to a further underestimation of the DOC photo remineralization.

Same comment as above.

References cited:

Cory, R. M., C. P. Ward, B. C. Crump, and G. W. Kling. Sunlight controls water column processing of carbon in arctic fresh waters. Science 345: 925–928. doi:10.1126/science.1253119, 2014.

Osburn, C. L., H. E. Zagarese, D. P. Morris, B. R. Hargreaves, and W. E. Cravero. Calculation of spectral weighting functions for the solar photobleaching of chromophoric dissolved organic matter in temperate lakes. Limnology and Oceanography 46: 1455–1467. doi:10.4319/lo.2001.46.6.1455, 2001.

---

## Author Response (AR3)

I made the following revisions in the included documents.

1) The author affiliation list was updated.
2) We posted our data to the Environmental Data Initiative.  I include the link in the appropriate section and included the citation in the reference section.

Please let me know if any additional information or changes are needed.